# Architectural Characteristics of Different Configurations Based on New Geometric Determinations for the Conoid

Joseph Cabeza-Lainez 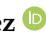

Department of Architectural Composition, University of Seville, Av. Reina Mercedes 2, 41002 Sevilla, Spain; crowley@us.es

**Abstract:** The aim of this article is to orient the evolution of new architectural forms offering up-to-date scientific support. Unlike the volume, the expression for the lateral area of a regular conoid has not yet been obtained by means of direct integration or a differential geometry procedure. In this type of ruled surface, the fundamental expressions I and II, for other curved figures have proved not solvable thus far. As this form is frequently used in architectural engineering, the inability to determine its surface area represents a serious hindrance to solving several problems that arise in radiative transfer, lighting and construction, to cite just a few. To address such drawback, we conceived a new approach that, in principle, consists in dividing the surface into infinitesimal elliptic strips of which the area can be obtained in an approximate fashion. The length of the ellipse is expressed with certain accuracy by means of Ramanujan's second formula. By integrating the so-found perimeter of the differential strips for the whole span of the conoid, an unexpected solution emerges through a newly found number that we call psi ($\psi$). In this complex process, projected shapes have been derived from an original closed form composed of two conoids and called Antisphera for its significant parallels with the sphere. The authors try to demonstrate that the properties of the new surfaces have relevant implications for technology, especially in building science and sustainability, under domains such as structures, radiation and acoustics. Fragments of the conoid have occasionally appeared in modern and contemporary architecture but this article discusses how its use had been discontinued, mainly due to the uncertainties that its construction posed. The new knowledge provided by the authors, including their own proposals, may help to revitalize and expand such interesting configurations in the search for a revolution of forms.

**Keywords:** conoid; ellipse; calculus of surface areas; number psi; number Pi; parametric design; cubature architecture; design paradigms

## 1. Introduction

*Outline of the Problem*

Since antiquity the meaning of the number Pi has been associated with the length of a circumference, that is, such length, if the diameter of the said circumference is the unit, equates Pi, and correspondingly for different measures of the diameter. Even in the Bible, when an injunction is transferred to building an offering's laver (the Sea Bronze) of circular design in Solomon's Temple, it is mentioned that the perimeter ought to be three times the diameter [1]—a revealing estimate.

Given this, it is reasonable to speculate, from a scientific point of view, about the meaning of different powers of Pi, for example, Pi$^N$?

The answer is positive, and in this article, we would discuss the particular case of the second power of $\pi$, that is, Pi squared or Pi$^2$. Such a situation arises when we try to calculate the surface area of a Conoid, a ruled surface generated by parallel straight lines that project from a circumference directrix onto a linear edge (Figure 1).

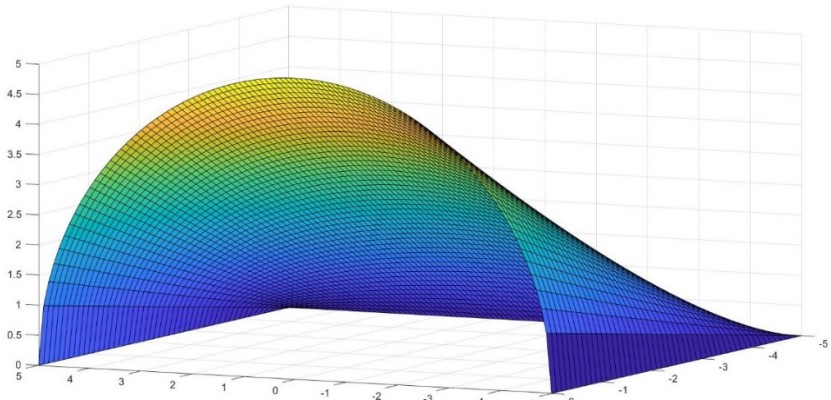

**Figure 1.** A typical straight conoid with circular directrix, where *R* = *L* = 5.

The equation that regulates such a warped figure for *x* and *y* positive is:

$$\frac{L^2 z^2}{(L-x)^2} + y^2 = R^2 \tag{1}$$

where *R* is the radius of the directrix in the case of a circumference and *L* is the length in the *X*-direction as shown in Figure 2. If we make *L* = *R*, it turns out that,

$$\frac{R^2 z^2}{(R-x)^2} + y^2 = R^2 \tag{2}$$

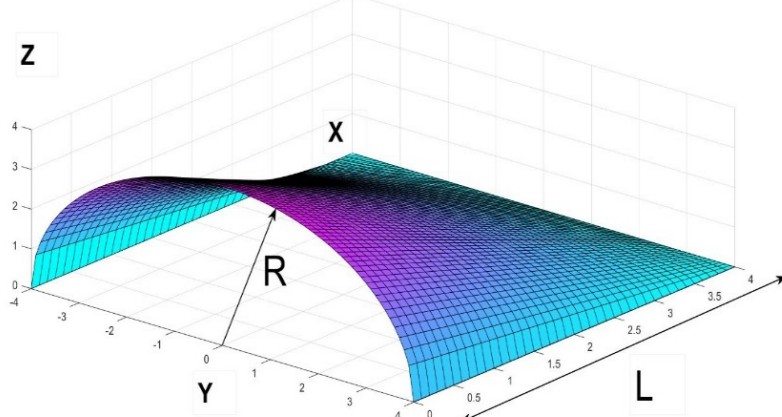

**Figure 2.** Explanation of the parts of a straight conoid with circular directrix, in this case *R* = *L* = 4.

To solve the problem of the lateral area of such surface, after failed attempts for a solution with other methods, we employ Ramanujan's second approximation for ellipses [2].

## 2. Methods and Materials

### 2.1. Resolution of the Proposed Approximation

In his well-known proposal to calculate the perimeter of an ellipse, Ramanujan stated that the perimeter (P) of the curve equates:

$$P = \pi \left( 3(a+b) - \sqrt{(3a+b)(3b+a)} \right) \tag{3}$$

where *a* is the major and *b* is the minor semi-axis. Such an approximation is fairly easy to handle and it works well for the extreme case of *b* = 0, a straight line, which tends to appear in the limit edge of the conoid and its boundaries.

The error committed by using this formula is not constant since it depends on the diverse conoidal sections. On the other hand, as we do not possess an exact expression for the length of an ellipse, there is no proper way to compute such error in a mathematical fashion. In Table 1 we have attempted to do so by numeric methods.

**Table 1.** Results of computing the area of several conoids by the method proposed by the author and with graphic interpolation procedures allowed by the software *Grasshopper* and the command *Alphashape* of Matlab.

| Radius = L Unit | Alpha-Shape | Cabeza Approx. | Grass-Hopper | Delta Δ Alphas | Delta Δ Grassh |
|---|---|---|---|---|---|
| 0.25 | 0.1958 | 0.1862 | 0.1842 | 0.0096 | −0.002 |
| 0.50 | 0.7712 | 0.7446 | 0.7369 | 0.0266 | −0.0077 |
| 1 | 3.0726 | 2.9784 | 2.9476 | 0.942 | −0.308 |
| 2 | 12.155 | 11.913 | 11.790 | 0.242 | −0.123 |
| 3 | 27.734 | 26.805 | 26.528 | 0.929 | −0.277 |
| 4 | 48.886 | 47.654 | 47.161 | 1.232 | −0.493 |
| 5 | 76.019 | 74.460 | 73.690 | 1.559 | −0.77 |
| 6 | 109.27 | 107.22 | 106.11 | 2.05 | −1.11 |
| 7 | 148.83 | 145.94 | 144.43 | 2.89 | −1.51 |
| 8 | 194.54 | 190.61 | 188.64 | 3.93 | −1.97 |
| 9 | 246.41 | 241.25 | 238.75 | 5.16 | −2.5 |
| 10 | 304.254 | 297.841 | 294.76 | 6.413 | −3.081 |

For the said directrix, we can substitute *a* for *R*, the radius of the circumference.

Thus, *a* = *R* and it is easy to prove that being *R/L*, the tangent of the angle formed by the middle section of the figure, the minor semi-axis is nothing but *b* = *x×R/L*. It should be noted that in the following calculations, the positive direction of the *x*-axis is reversed for simplicity from that in Figure 2 and the origin of coordinates is at the linear end of the figure and not at the extreme of the circumference.

Then, as previously stated, *R* is the radius of the end semicircle and *L* is the total length and for any *x* value between 0 and *L*, we would obtain,

$$P = \pi \left( 3(R + xRL) - \sqrt{ \left( 3R + \frac{xR}{L} \right) \left( \frac{3xR}{L} + R \right) } \right) \tag{4}$$

And grouping similar terms,

$$P = \pi \left( 3R(1 + x/L) - \sqrt{R(3 + x/L)R(3x/L + 1)} \right) \tag{5}$$

$$P = \pi \left( 3R(1 + x/L) - \sqrt{R^2(3 + x/L)(3x/L + 1)} \right) \tag{6}$$

The half perimeter is,

$$P_1 = (\pi/2) \left( 3R(1 + x/L) - \sqrt{R^2(3 + x/L)(3x/L + 1)} \right) \tag{7}$$

We can take *R* out of the whole expression,

$$P_1 = (\pi R/2) \left( 3(1 + x/L) - \sqrt{(3 + x/L)(3x/L + 1)} \right) \tag{8}$$

And *L* as well,

$$P_1 = (\pi R/(2L)) \left( 3(L + x) - \sqrt{3x^2 + 10Lx + 3L^2} \right) \tag{9}$$

To obtain the lateral area of the conoid composed by diminishing strips along the central section, we need to perform the integration of,

$$A_c = \frac{\pi R}{2L} \int_0^L \left[ 3L + 3x - \sqrt{3x^2 + 10Lx + 3L^2)} \right] dx \tag{10}$$

The first two terms are immediate and give the solution;

$$I_1 + I_2 = \left[ 3Lx + \frac{3x^2}{2} \right]_0^L \tag{11}$$

And the final result for these two terms, applying the limits of integration is,

$$I_1 + I_2 = 3 \times L^2 + \frac{3L^2}{2} = \frac{9L^2}{2} \tag{12}$$

This, multiplied by the constants out of the integral gives, $\frac{9\pi RL}{4}$;

The third term is slightly more complicated following the square root type and it involves a logarithmic primitive.

$$I_3 = \int_0^L \left[ \sqrt{3x^2 + 10Lx + 3L^2} \right] dx \tag{13}$$

This integral presents the root of an expression of the type $a + bx + cx^2$.

As such, it is recommended to find the value of $\Delta = 4ac - b^2$, for this case, $36L^2 - 100\,L^2 = -64L^2$

The solution yields, [3]

$$I_3 = \left[ \frac{6x + 10L}{12} \sqrt{3x^2 + 10Lx + 3L^2} - \frac{64L^2}{24} \left( \frac{1}{\sqrt{3}} \right) \log \left( 2\sqrt{3(3x^2 + 10Lx + 3L^2)} + 6x + 10L \right) \right]_0^L \tag{14}$$

And substituting,

$$I_3 = \left[ \frac{4L}{3}(4L) - \frac{64L^2}{24} \left( \frac{1}{\sqrt{3}} \right) \log \left( 2\sqrt{3(16L^2)} + 16L \right) \right] - \left[ \frac{10L}{12} \sqrt{3L^2} - \frac{64L^2}{24} \left( \frac{1}{\sqrt{3}} \right) \log \left( 2\sqrt{3(3L^2)} + 10L \right) \right] \tag{15}$$

$$I_3 = \left[ \frac{16L^2}{3} - \frac{5\sqrt{3}L^2}{6} + \left( \frac{8L^2}{3\sqrt{3}} \right) \log(16L) - \left( \frac{8L^2}{3\sqrt{3}} \right) \log \left( 8L\sqrt{(3)} + 16L \right) \right] \tag{16}$$

By virtue of the properties of division of the logarithm,

$$I_3 = L^2 \left[ \frac{16}{3} - \frac{5\sqrt{3}}{6} - \left( \frac{8}{3\sqrt{3}} \right) \log \left( \frac{\sqrt{3} + 2}{2} \right) \right] \tag{17}$$

And from Equation (12), the sum of the two previous immediate integrals was,

$$I_1 + I_2 = \frac{9L^2}{2}$$

The total result for the so-conceived area, subtracting $I_3$ from Equation (17) is,

$$\frac{\pi R}{2L} L^2 \left[ \frac{9}{2} - \frac{16}{3} + \frac{5\sqrt{3}}{6} + \left( \frac{8}{3\sqrt{3}} \right) \log \left( \frac{\sqrt{3} + 2}{2} \right) \right] \tag{18}$$

After careful simplification, the Area of the conoid based in *R* and *L* gives:

$$I = \frac{\pi RL}{2}\left[\frac{27-32}{6} + \frac{5\sqrt{3}}{6} + \left(\frac{8\sqrt{3}}{9}\right)\log\left(\frac{\sqrt{3}+2}{2}\right)\right] \tag{19}$$

which can be reduced to,

$$I = A = \frac{\pi RL}{2}\left[\frac{5\sqrt{3}}{6} - \frac{5}{6} + \left(\frac{8\sqrt{3}}{9}\right)\log\left(\frac{\sqrt{3}+2}{2}\right)\right] \tag{20}$$

And consequently,

$$A = \frac{\pi RL}{4}\left(\frac{1}{9}\left[15\left(\sqrt{3}-1\right) + 16\sqrt{3}\,\log\left(\frac{\sqrt{3}+2}{2}\right)\right]\right) \tag{21}$$

### 2.2. Discussion of the Findings

For several reasons we have decided to name this new number $\psi$,

$$\psi = \frac{1}{3^2}\left[15\left(\sqrt{3}-1\right) + \left(4^2\right)\sqrt{3}\,\log\left(1+\frac{\sqrt{3}}{2}\right)\right] = 3.140923532703498 \tag{22}$$

$$\psi \cong \pi \tag{23}$$

For computational purposes both numbers can be equated, and it will considerably simplify Equation (21) as,

$$A = \frac{\pi RL}{4}[\psi \cong \pi] = \frac{\pi^2 RL}{4} \tag{24}$$

We find such a result remarkable in sundry senses. Firstly, because, $\psi$ is a transcendental number [4], akin to Pi but original in concept. Secondly, if substituted for Pi in the discussion it could perhaps improve the accuracy of Ramanujan's approach, and this remains a question open for discussion in future developments.

Thirdly, it involves the fact that the area of the figure studied, the conoid, if *R* = *L* = 1 would be a fourth of Pi squared or Pi multiplied by itself.

### 2.3. Definition of the Antisphera

A volume composed of four symmetrical opposed conoids, as previously defined (radius = 1), would have a lateral surface of $\pi^2$, that is Pi squared.

When conceiving such a form, obvious similarities with the sphere come to mind and that it is why we have named this curious figure in Grecian "*Antisphera*". The plan of the figure resembles a square of two by two, but the front view is a circle of diameter one. It constitutes a three-dimensional example of the circle's quadrature, sought since antiquity [5].

Its equation responds to,

$$\frac{R^2 z^2}{(\pm(R-x))^2} + y^2 = R^2 \tag{25}$$

We consider just the former to be a noticeable finding. However, the problem is not completely solved since the parallel infinitesimal strips may not fit perfectly with the curved surface and they would act as a kind of envelope. For engineering purposes this is deemed sufficient when the angle of inclination of the figure, $\theta$, is not too steep (less than $\pi/2.5$), but in order to improve accuracy, we have calculated a coefficient to take into account the differences of width of the said strips.

Since we certainly know that at the middle section of the surface, the width of an actual unit section of the conoid is not $dx$ but instead $dx/\cos\theta$, we need to take into account this feature and effect an interpolation between both values.

It can be discussed whether the variation of the strip is curvilinear or not, but several experimental calculations (See Section 2.4), estimate that for the time being, this is the preferred approximation for angles under Pi/2.5 (other approaches are also being considered for higher values of $\theta$, see Section 2.5).

Such a coefficient of Cabeza-Lainez, $\kappa$, would accordingly be,

$$\kappa = \frac{1 + \sec\theta}{2} = \frac{1 + \cos\theta}{2\cos\theta} \tag{26}$$

where $\tan(\theta) = R/L$ as $\theta$ is the arctangent of the central section of the figure.

Therefore, the final expression to compute the lateral area of this particular surface, with the caveats referred above, produces,

$$A = \frac{\pi^2 RL(1 + \cos\theta)}{4} \frac{}{2\cos\theta} \tag{27}$$

But if $R$ and $L$ coincide, Equation (27) is reduced to,

$$A = \frac{\pi^2 R^2}{4} \frac{\left(1 + \sqrt{2}\right)}{2} \tag{28}$$

In the case of a new volume that we have defined as *Antisphera* ©, if R is the unit and if we adjoin the four quarters of which it is composed (Figure 3),

$$A = \pi^2 R^2 \frac{\left(1 + \sqrt{2}\right)}{2} = \pi^2 \frac{\left(1 + \sqrt{2}\right)}{2} \tag{29}$$

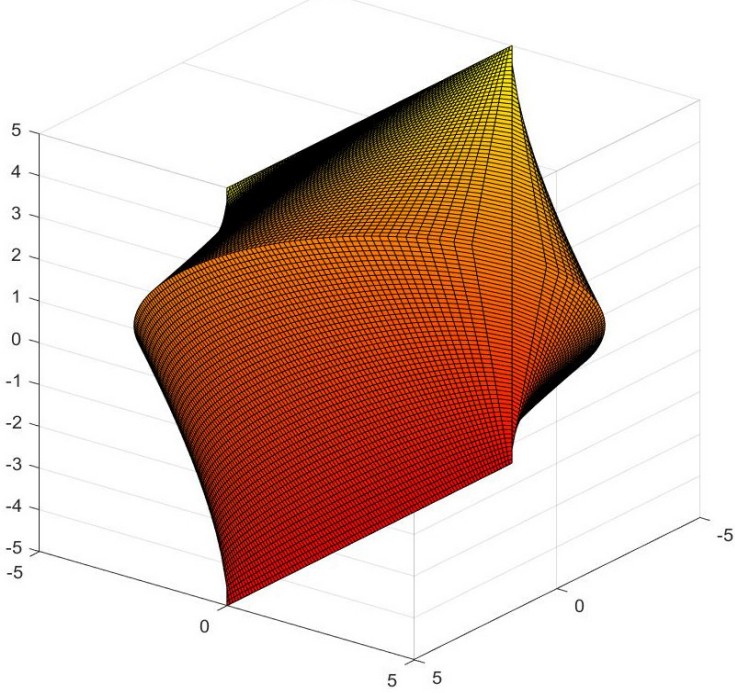

**Figure 3.** Depiction of the *Antisphera* for *R* = 5. View from below.

Nevertheless, if we should make,

$$R^2 = \frac{2}{\left(1 + \sqrt{2}\right)} \tag{30}$$

It would mean that for an *Antisphera* of precise Radius = 0.9102,

$$A = \pi^2 \tag{31}$$

In this fashion, a new and distinct meaning, has been attributed to the second power of $\pi$, by virtue of such elaborate demonstration.

### 2.4. Comparison with Other Approximate Computing Methods

Subsequently, we have proceeded to compare the aftermath with other numerical simulations available, for instance the command *Alphashape* for computation of areas in Matlab and the graphic interface *Grasshopper*. The results show a considerable agreement, and our findings stay in the middle of the output for both approximate tools (Table 1).

More thorough data are being prepared, but the procedure for the graphic interfaces is clumsy and haphazard as a bespoke volume is required every time in order to compute the areas, and the mesh of interpolation has to be decided beforehand with frequent hollow regions. This is a clear advantage of our method.

### 2.5. Discussion for Higher Values of the Conoid Angle

The $\kappa$ coefficient previously defined must, for coherence, remain over,

$$\kappa > \frac{2 \tan \theta}{\pi} \tag{32}$$

It is however clear that, for values of $\theta > 1.25$ the former relation may cease to verify (Figure 4).

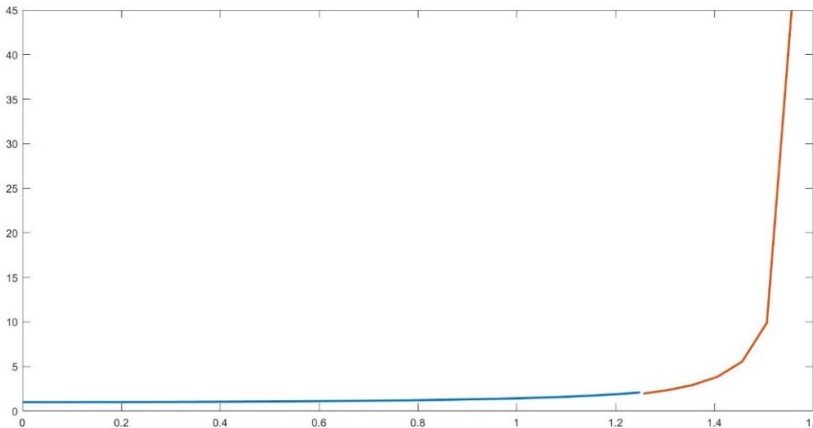

**Figure 4.** Depiction of Cabeza-Lainez coefficient $\kappa$ (blue) and limit case (red).

For lower quantities, the coefficient performs smoothly, since $\cos \theta$ tends to be one and the same as $\kappa$ does, so our prediction becomes even more accurate.

But for values over Pi/2.5, it is convenient to define a new coefficient. We are currently estimating different adjustments to present them in a further development of our theory. In the meantime, a tentative assumption for the value of the area, would be the following,

$$A = \frac{\pi^2 RL}{4} + \frac{\pi R^2}{2} = \frac{\pi R}{4}(\pi L + 2R) = \frac{\pi^2 RL}{4}\left(1 + \frac{2R}{\pi L}\right) \tag{33}$$

Thus, the new coefficient $\kappa_1$ would be,

$$\kappa_1 = 1 + \frac{2 \, \text{tg} \, \theta}{\pi} \tag{34}$$

A suggested experimental refinement of this factor from $\theta = \pi/2.5$ and onwards is,

$$\kappa_2 = \sin\left(\frac{\pi}{2} - \theta\right) + \frac{2 \, \text{tg}\, \theta \, \sin \, \theta}{\pi} \tag{35}$$

In most problems of engineering, high values of the said angle are rare because they imply that the conoid is very close to the circle or in other words, there is little or no space left inside the surface, which may become contradictory to the nature of spatial design [6].

*2.6. Calculations of the Area for an Elliptic Conoid*

In Section 2.1 we had delayed the discussion after attaining the area values for the conoid ending in a circumference, but in a similar manner we can continue to use Ramanujan's prediction for an ordinate fragment of the same conoid whose extreme is logically an ellipse. For any real number $n$, the area yields,

$$A_{ec} = \frac{\pi R}{2L} \frac{(1 + \cos \, \theta)}{2 \cos \, \theta} \int_0^{nL} \left[3L + 3x - \sqrt{3x^2 + 10Lx + 3L^2}\right] dx \tag{36}$$

$$A_{ec} = \frac{\pi RL}{216} \frac{(1 + \cos \, \theta)}{2 \cos \, \theta} \left[\begin{array}{c} 9\left(18n(2+n) - (6n+10)\sqrt{n(3n+10)+3} + 10\sqrt{3}\right) + \\ 96\sqrt{3} \, \log\left(\frac{\sqrt{3n(3n+10)+9} + 3n + 5}{8}\right) \end{array}\right] \tag{37}$$

For $n = 1$ the solution is Equation (21).

As before, we have employed the coefficient $\kappa$, but we need to be aware that for values of $\theta$ nearing $\pi/2.5$ the expressions proposed in the previous section, that is, $\kappa_1$ or $\kappa_2$ should be introduced in its stead.

*2.7. Calculations of the Volume of the Conoid*

As a complement for the theories exposed, the computation of the volume of the conoidal figure by means of the previous integral method is relatively simple and exact. Since the value of the area of an ellipse is $\pi ab$, and as before, $a = R$ and $b = xR/L$

$$V = \pi R \int_0^{nL} [xR/L] dx = \frac{\pi R^2 n^2 L}{2} \tag{38}$$

This equates the volume of the equivalent cylinder multiplied by s = $n/2$.

If $n = 1$ and the limit is $L$,

$$V = \frac{\pi R^2 L}{2} \tag{39}$$

that is half the volume of the equivalent cylinder (as s = ½).

Finally, if $L = R$, the volume gives

$$V = \frac{\pi R^3}{2} \tag{40}$$

Making $R = 1$ as in the *Antisphera*, we receive $V = \pi/2$.

With $R = 0.9102$ as in Equation (30), the volume reaches, 0.3770 $\pi$.

The volume and area properties in the conoid are smaller than in the cylinder but still larger than the equivalent cone. This will prove advantageous for the sustainability of structures and buildings (Sections 5 and 6) as the envelope and consequently the energy exchange and materials are less costly.

The finding of the volume of a conoid is sometimes attributed to Johannes Kepler but to our knowledge the first polymath to deduct it by comparison with the volume of a cone of the same basis, was Guarini in *Euclides Adauctus* [5].

### 3. Repercussions for Radiative Heat Transfer

*3.1. Introduction to the Problem of Surface Factors*

In previous studies [6], some of which were published in this journal [7], Cabeza-Lainez highlighted that the radiative exchange factor for any manifold surface is dependent on the equation:

$$F_{12} = \frac{1}{A_1} \left[ \int_{A_2} \int_{A_1} \frac{\cos \theta_1 \cos \theta_2}{\pi r^2} dA_1 dA_2 \right] \tag{41}$$

In a volume composed of only two surfaces, the same author established its second principle of radiation [8], which states that in the said situation, the rate of exchange is proportional to the areas of the intervening surfaces. For example, in a hemisphere (see Figure 5), the respective areas are $A_1 = 2\pi R^2$ and $A_2 = \pi R^2$.

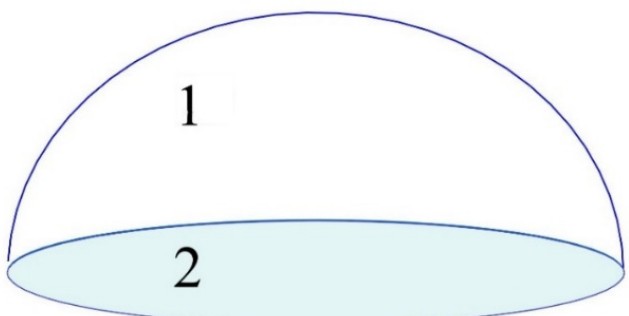

**Figure 5.** The second radiation principle of Cabeza-Lainez form-factors applied to a hemisphere.

As such, the corresponding form-factor from the half sphere to its base disc is [8], $F_{12} = \frac{A_2}{A_1} = \frac{1}{2}$, and the amount of energy from the hemisphere to itself, $F_{11}$ is also ½ [8].

By the principle of conservation of energy, all interchanges must add up to 100% or unity [6].

However, if the disc were topped by a double symmetric conoid (Figure 6), since previous to our finding we ignored the value of its lateral area, the question remained unknown.

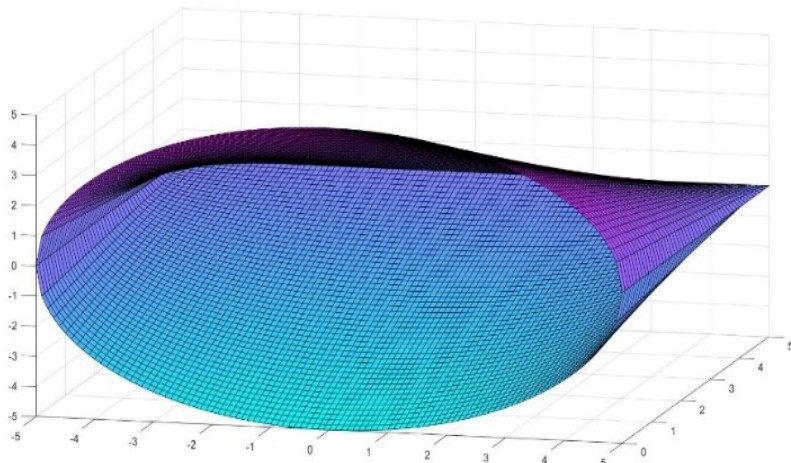

**Figure 6.** The second radiation principle of Cabeza-Lainez applied to the form-factor of a double conoid.

We are finally in the position to respond to this issue with perfect ease. It is opportune to outline that the above integral equation, Equation (41), is deemed unsolvable for the conoidal geometry [7].

The area of this double conoid is,

$$A = \frac{\pi^2 R^2}{2} \frac{\left(1 + \sqrt{2}\right)}{2} \tag{42}$$

*3.2. Example 1*

In all the cases where $\theta = \pi/4$, the relationship between the area of the base circle $\pi R^2$ and the double conoid is precisely,

$$F_{12} = \frac{A_2}{A_1} = \frac{4}{\pi\left(1 + \sqrt{2}\right)} = 0.527393 \tag{43}$$

This is the factor from the conoidal top to the circular base and since $F_{11} + F_{12} = 1$, the self-factor $F_{11}$ is then,

$$F_{11} = 1 - F_{12} = 1 - 0.527393 = 0.4726 \tag{44}$$

For this particular disposition, the values are not dissimilar from those of the hemisphere, 0.5, but they will need to be included in a standard comparison among radiative shapes described in reference [8].

Let us now discuss a new and more difficult situation for a conoid with radius 2 and length 4 (Figure 7).

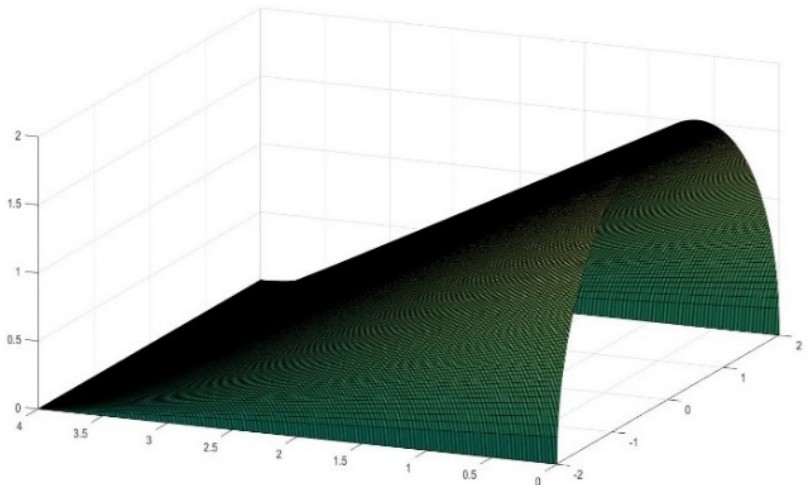

**Figure 7.** The single conoid limited by a rectangular plane and semicircle.

*3.3. Example 2*

In reference [9], Cabeza-Lainez has partly solved Equation (31) for some circular fragments.

For any vertical circular sector with a center situated in the middle of the edge $x$ of a horizontal rectangle of dimensions, $x$ and $y$; by virtue of the Cabeza-Lainez seventh principle, [10] the configuration factor from the sector of radius $r$ to a point in the perpendicular rectangle, will be:

For $t = r^2 + y^2 + x^2$, $m = \sqrt{x^2 \sin^2 \theta_1 + y^2}$ and $n = \sqrt{x^2 \sin^2 \theta_2 + y^2}$

$$f_{12} = \frac{y}{2\pi} \times \left( \frac{\cos \theta_1}{m} \arctan \frac{r}{m + \frac{\cos \theta_1 x}{m}(\cos \theta_1 x - r)} - \frac{\cos \theta_2}{n} \arctan \frac{r}{n + \frac{\cos \theta_2 x}{n}(\cos \theta_2 x - r)} \right) + \frac{y}{4\pi x} \ln\left[\frac{(t - 2 \cos \theta_1 rx)}{(t - 2 \cos \theta_2 rx)}\right] \tag{45}$$

Bearing in mind that the sector is comprised between the angles $\theta_2$ and $\theta_1$ and being its radius $r$ as mentioned.

In the usual situation of a semicircle the above expression is reduced to,

$$f_{12} = \frac{1}{2\pi} \left( \arctan \frac{r+x}{y} + \arctan \frac{r-x}{y} \right) + \frac{y}{4\pi x} [\ln \left( r^2 + y^2 + x^2 - 2rx \right) - \ln(r^2 + y^2 + x^2 + 2rx)] \quad (46)$$

By numerical procedures detailed in [7], we extend the above expression to the whole rectangle to find the form-factor, whose value is of $F_{12} = 0.1272$. It represents fraction of exchange of radiative energy from the horizontal rectangle under the conoid ($A_1$) to the semi-circular side of the figure ($A_2$).

The previously unknown area of the conoid $A_3$, following the above stated formulas (Equation (27)) is 20.9042

From the reciprocity principle [6], $A_1 F_{12} = A_2 F_{21}$, whence,

$$F_{21} = (16/2\pi)0.1272 = 0.3239 \quad (47)$$

Being $A_1$ and $A_2$ planar, it is mandatory that [6],

$$F_{12} + F_{13} = 1 \quad (48)$$

$$F_{21} + F_{23} = 1 \quad (49)$$

And this implies,

$$F_{13} = 1 - 0.1272 = 0.8728 \quad (50)$$

$$F_{23} = 1 - 0.3239 = 0.6761 \quad (51)$$

Applying reciprocity again [6], $A_3 F_{31} = A_1 F_{13}$ and $A_3 F_{32} = A_2 F_{23}$, which yields,

$$F_{31} = (A_1/A_3) F_{13} = (16/20.9042) \, 0.8728 = 0.6680 \quad (52)$$

$$F_{32} = (A_2/A_3) F_{23} = (2\pi/20.9042) \, 0.6761 = 0.2032 \quad (53)$$

By virtue of the principle of conservation of energy [6],

$$F_{31} + F_{32} + F_{33} = 1 \quad (54)$$

$$F_{33} = 1 - F_{31} - F_{32} \quad (55)$$

Being non-planar, the fraction of energy that the radiating conoid exchanges with itself is,

$$F_{33} = 1 - 0.6680 - 0.2032 = 0.1288 \quad (56)$$

Not merely radiative heat transfer in the figure under study has been solved by this procedure, but also light transmission when it originates at conoidal skylights such as those constructed by Ilja Doganoff [11] in 1957 in Bulgaria (See Section 5.1).

### 3.4. Example 3

If we, as in a sort of check, would double the conoid presented in example 2 and compare in it the factor between a circle ($A_2$) and ($A_1$) the enclosing figure, (Figure 8) we obtain that, since the relation of areas is now 0.3006, the factor from the conoid to the circle is precisely this value, following Cabeza-Lainez' second principle [8].

Accordingly, the complex self-factor of the conoid to itself yields nothing but,

$$F_{11} = 1 - 0.3006 = 0.6994 \quad (57)$$

In the original half *Antisphera*, if we remember Section 3.2, and example 1, the same quantity amounted to,

$$F_{11} = 1 - 0.527393 = 0.4726 \quad (58)$$

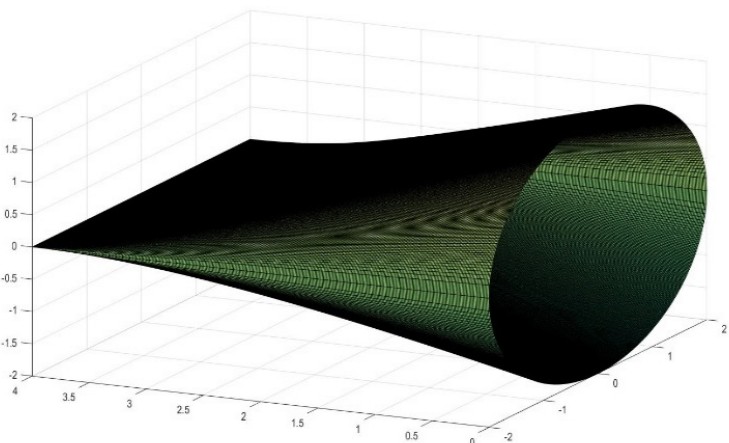

**Figure 8.** The same conoid of Figure 7 but closed to obtain the form-factor.

Such a difference can be explained because the area of this new conoid is larger than the previous one since the angle θ is less pronounced (in this case, the length doubles the radius), or in other words more energy is retained under the new configuration.

In the second example discussed, of a half-conoid (Section 3.3) involving three surfaces (rectangle, semicircle and conoid), the self-factor (Figure 9) was smaller, 0.1288, but a noticeable fact is that even so, it does not reach a fraction of one half of the self-factor for the whole conoid (0.6994/2 = 0.3497) of the same dimensions (Example 3), as logic would perhaps induce us to think. This exemplifies the complexity of the solutions because they often appear to rule out common sense [12].

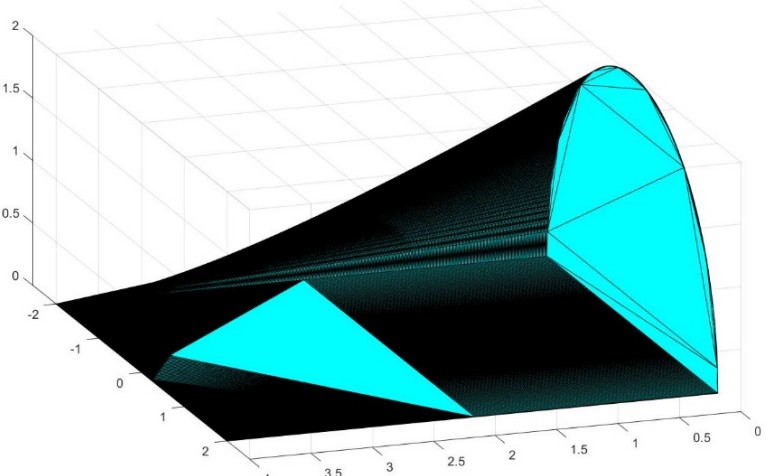

**Figure 9.** The three surfaces intervening in Example 2 whose area is calculated by computer graphic interpolation.

It is important to stress the utmost difficulty of obtaining these entities by any other method, including quadruple integration [7].

## 4. Generation of New Figures Based on the Previous Findings

In Section 2, we defined the symmetrical figure composed of four conoids, namely the *Antisphera* (Figure 10).

Nevertheless, we have conceived that by altering the symmetry and parts of the previous figure, a series of other interesting bodies is derived, maintaining initially the ratio *R* = *L*.

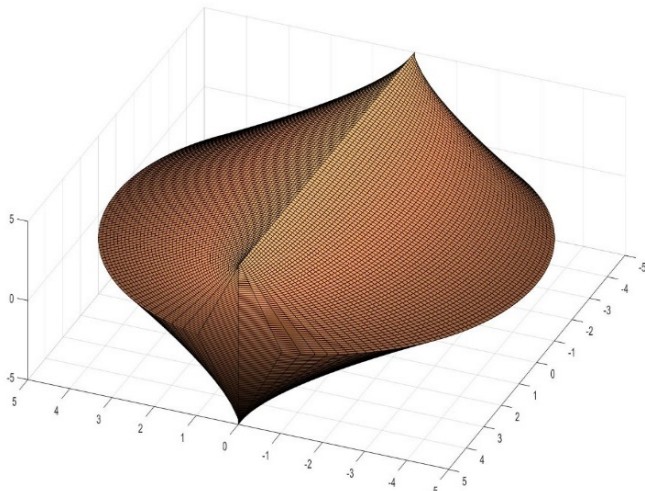

**Figure 10.** Three-dimensional rendering of the *Antisphera*. View from above.

The first one is opposed geometrically to the *Antisphera* because the edge straight lines intersect at its center plane, which is void. Due to this elusive, and somewhat dual, nature we have coined the name *Dyosphera* © for this shape (Figure 11).

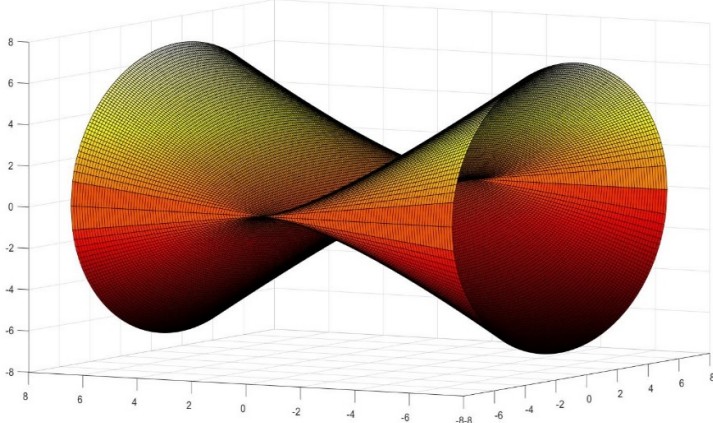

**Figure 11.** The horizontal part of the body known as *Dyosphera*.

In total, the *Dyosphera* features eight conoidal sections, organized in groups of four, rotated $\pi/2$ degrees (Figure 12)

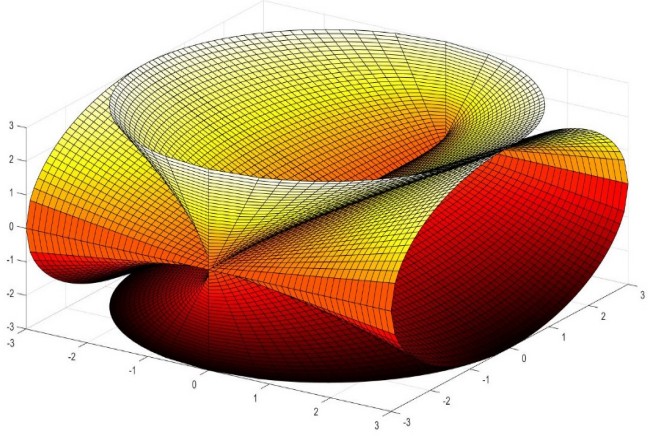

**Figure 12.** Distorted depiction of the complete *Dyosphera*.

The corresponding Equations are,

$$\frac{R^2 z^2}{x^2} + y^2 = R^2 \tag{59}$$

Combined with,

$$\frac{R^2 x^2}{z^2} + y^2 = R^2 \tag{60}$$

The open cavities and sinuous receptacles of this figure make it particularly suitable for aeronautical and machinery parts. It is also a very stable form because it has four circular bases.

The horizontal parts of the *Dyosphera* can be adroitly combined with the upper part of the *Antisphera* (Figure 13), to obtain a different figure, which we have called *Alosphera* ©.

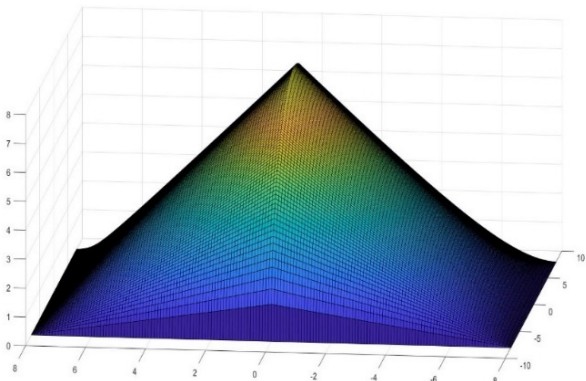

**Figure 13.** Detail of the upper part of the *Antisphera*.

In Figure 14, we see the Alosphera represented; its main feature is that two of the bases are square and two are circular, making it suitable for different uses, with straightforward storage of various units; we have come to guess that some cellular growths of different organisms may respond to this evolutive pattern.

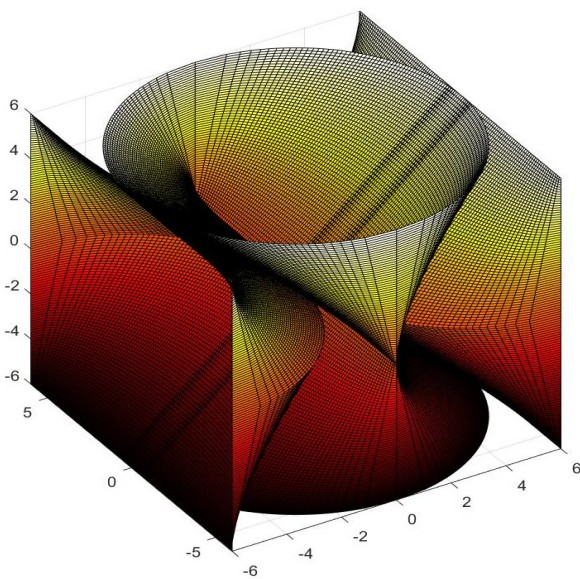

**Figure 14.** The Alopshera.

The *Alosphera* is the first Antisymmetric figure that we have identified but, similar to the *Dyosphera*, it also presents eight conoids.

Finally, a very important antisymmetric finding is the *Pterasphera* (Figure 15).

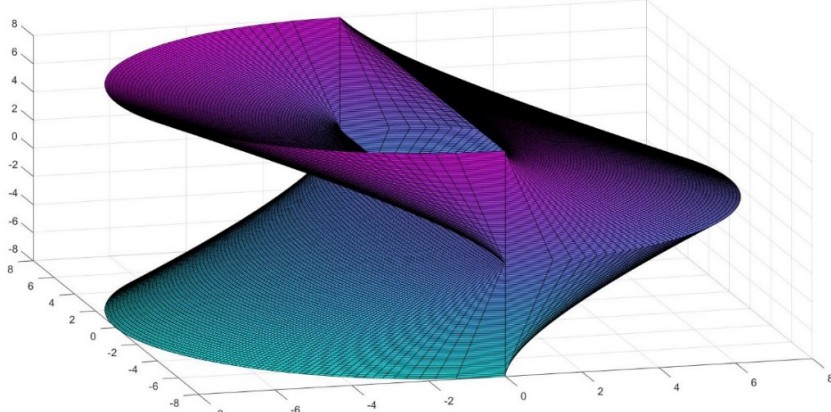

**Figure 15.** Initial Section of the *Pterasphera*.

This body as we shall discuss in Section 6 is tubular in nature. Being internally connected in its entirety, it is apt for conducting all kinds of fluids in an advantageous manner since, for instance, it can reduce the velocity and, at the same time, the noise of transporting the required fluids. Unlike *Antisphera*, it is self-standing and well balanced, which renders it suitable for elongation in the manner of a tower.

## 5. Architectural and Engineering Significance of the Geometric Findings

### 5.1. Historical Evolution

To our knowledge the first mathematician to introduce the conoidal figure was C. Guarino Guarini [5] in his famous treatise *Euclides Adauctus et Methodicus*. (Figure 16)

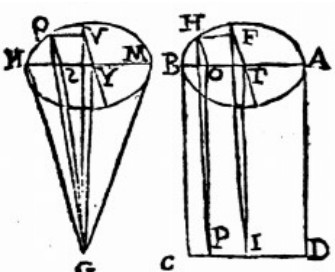

**Figure 16.** The point-cone and the cone that ends in a line according to Guarini.

He claims that only he has discovered the form (Figure 17) and says that it is a cone that ends in a straight line. (Later he uses the word hyperbolic conoid but for an entirely different body [5]).

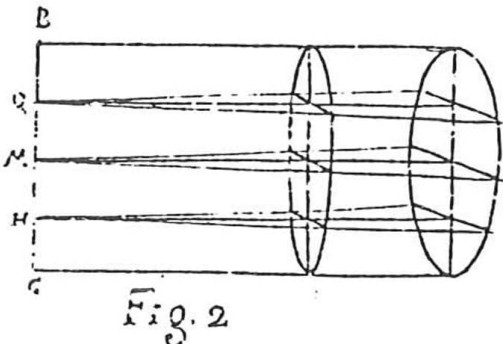

**Figure 17.** *Architettura Civile*. Camillo G. Guarini. Lastra IX. Trat. IV. Depiction of a cone ending in a line.

In another monumental work entitled *Architettura Civile*, published posthumously in 1737, he again refers to this unusual cone, stating that it has limited applicability in the corners of chamber vaults; we surmise that as a kind of squinch. He goes so far as to calculate the volume of the figure accurately, in the sense that we have explained in Section 2.7.

After this, the form remains all but forgotten in Architecture until, at the beginning of the 20th century, it receives a definitive impulse by the hand of revival architects such as Antonio Gaudi, who designed a roof with serially alternated conoids for a school building in front of his Sagrada Familia Cathedral in Barcelona (Figure 18).

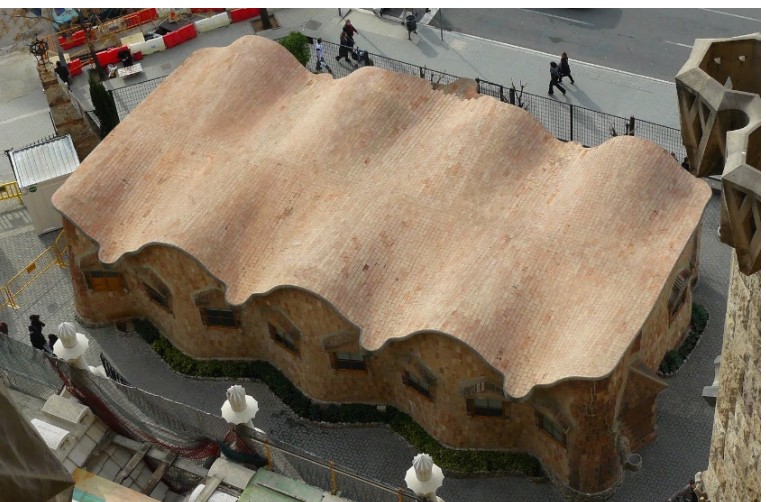

**Figure 18.** The Sagrada Familia Schools re-built. A. Gaudi.

Around the 1930s, the advent of concrete shell construction favored a revolution of engineering forms and the conoid surface was the recipient of much interest, especially for building hangars, factories and warehouses [13].

One of the best extant examples is the work of the Bulgarian Engineer Ilja Doganoff who, in 1956–1957, erected a repair workshop for Bulgarian Railways featuring a hundred conoid skylights (Figures 19–21) .

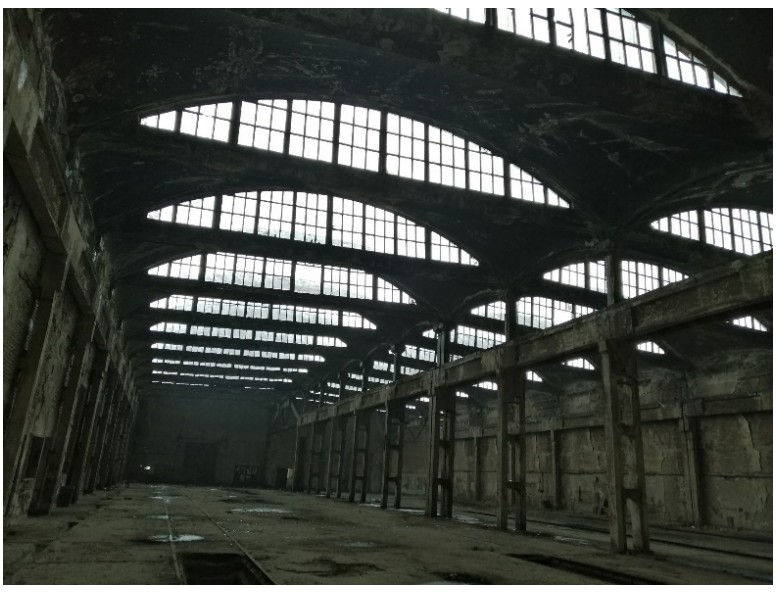

**Figure 19.** Ilja Doganoff. Current state of the Railway Depot. Source: Author.

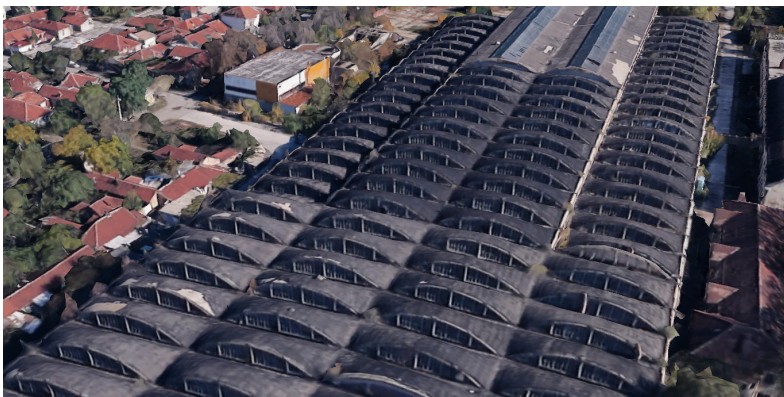

**Figure 20.** Recent aerial view of the Railway Depot by I. Doganoff.

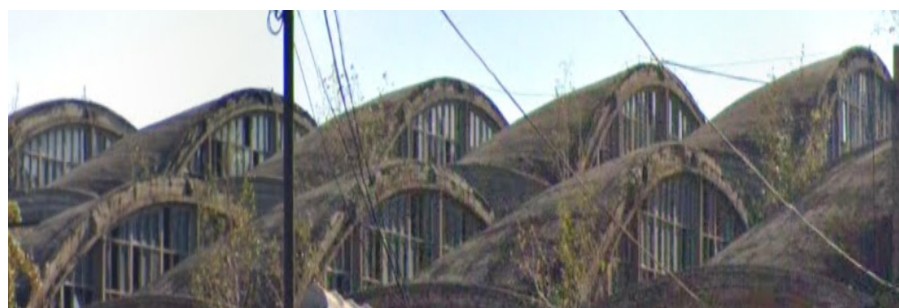

**Figure 21.** Close-up of the array of the conoids still standing after 64 years without maintenance.

The shapes were prefabricated in situ and then put on the roof with the help of a crane. They are an example of the extreme feasibility of the surfaces. We have calculated the daylighting transmission of similar shapes in Section 3, an undertaking neither Doganoff nor Ramaswamy [13] were able to perform.

Ramasamy [13] and Doganoff [11,14] report that owing to the want of knowledge about the surface, structural calculations turn out to be cumbersome, yet engineers still cherish the form because of its many advantageous properties and elegance, citing lighting and economy of construction [15,16] as potential reasons to explain their predilection.

### 5.2. Recent Projects of Conoids Realized by J. M. Cabeza-Lainez

The present author has been working in such conoid shapes for more than twenty-five years and his experience has ignited in part the present article. Modeling of the characteristic structural, acoustic and lighting properties of conoids has encompassed a significant amount of my career as researcher. Based on that, I can attest to its sustainability and endurance [17].

From the structural point of view, as a ruled surface, it can be built directly through straight lines (beams or poles); this fact greatly facilitates the construction and scaffolding, as more natural materials such as bricks or bamboo rods can be used without difficulty, even for reinforcement or repair.

Carbon-fiber coating has become a recent alternative for reinforcement, although moderately expensive.

The arched section of the conoid, whether circular or elliptic, presents a vertical tangent, shown in the previous sections [18]. Therefore, if adequately constructed, it is free from horizontal thrusts that might compromise the supporting frame. In other words, it transmits all the loads of the structure vertically and avoids the use of buttresses (Figure 22).

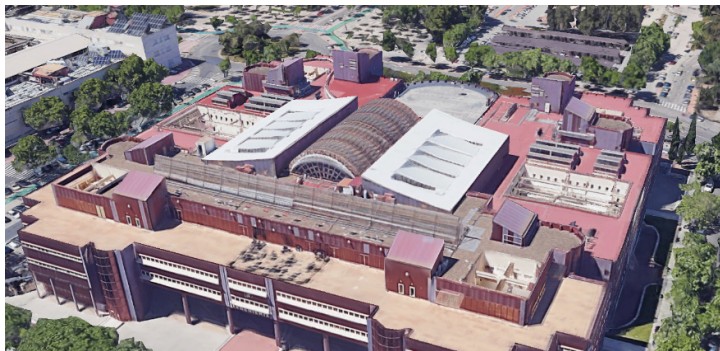

**Figure 22.** School of Engineering of the University of Sevilla. Aerial view of conoidal skylights.

These consistent and diminishing arches function as girths for most parts of the surface [19] and provide increased resistance to a significant degree. It is true that calculation of hyper-static arches is not widely treated in the literature, but we suggest the column analogy method proposed by H. Cross [20] as a helpful and programming-friendly procedure.

Due to its curvature, the aerodynamics of the roof is excellent for bearing wind loads and other meteorological phenomena such as rain, drizzle or snow. At the same time, because of the former, it enhances air flow from the outside or from the internal stack effect with appropriate vents.

Regarding lighting properties if, as usual, the glazed apertures lie in the curvilinear extremes of the forms, they bring uniform illuminance, as we calculated in Section 3 (Figure 23) and can be easily shaded by eaves protruding from the same brim of the surface (Figure 24).

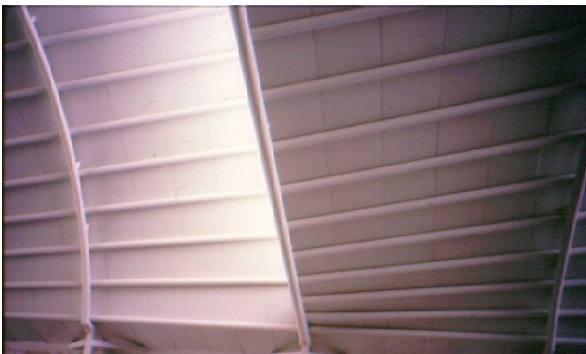

**Figure 23.** Interior view showing light diffusion at the central conoid. School of engineering of Seville.

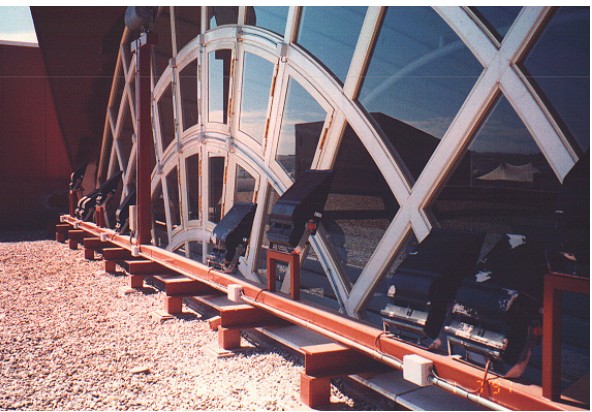

**Figure 24.** Semicircular opening and projecting overhang at the extreme of the central conoid of the School of Engineering of Seville.

Acoustic properties stem from the circumstance that the inside surface of the conoid is mostly convex as we have checked mathematically [6]. Sound waves are diffused in this kind of ceiling and consequently, noise and reverberation become dampened. If, through appropriate design, the conoid covers a trapeze or fan-shaped plan (Figures 25 and 26) the effect of an even sound pressure is manifest [21]. (In this last case the surface is not a proper conoid as the forming lines are not all parallel to a common plane).

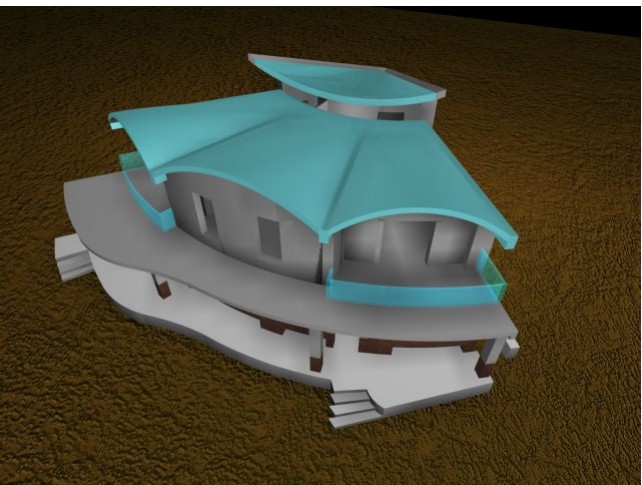

**Figure 25.** Three rotated conoidal roofs designed for a musicians' family. Sanlucar (Seville).

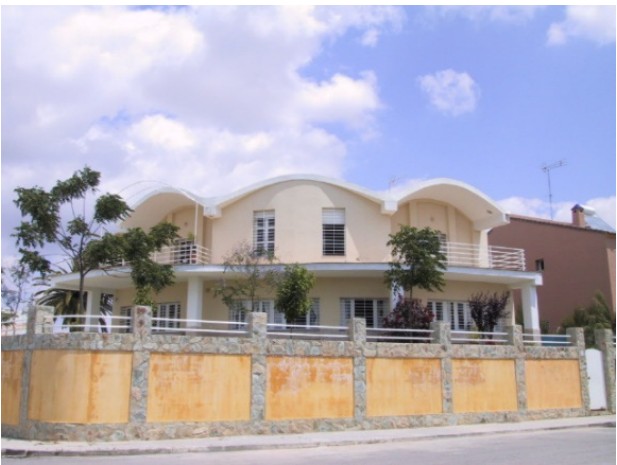

**Figure 26.** View of the house of the musicians. The vaults are entirely constructed in brick with occasional steel reinforcement.

The aforementioned acoustic benefits are extensive to interior illumination for the same reason of convexity of forms.

The cover in Figure 25, strictly-speaking is not a conoid, because its equation differs from what we have explained above. The forming straight lines are not parallel to a plane but they all coincide along a central vertical axis; such is a new form named Cabeza-Abajo surface (after the author). However, topping of a fan-shaped plan with this kind of surface offers a very interesting structural property: the larger spans between pillars are covered by arches, while the tapered end of the trapeze features a common slab or planar beam, which seems very logical from the constructive aspect [21].

In this way, the shells' materials can be lighter and smoother, in Figure 27 we present three vaults consisting of thin layers of hollow brick with steel mesh as a reinforcement. The result has proven to provide increased insulation and adds variety of light effects (Figure 28).

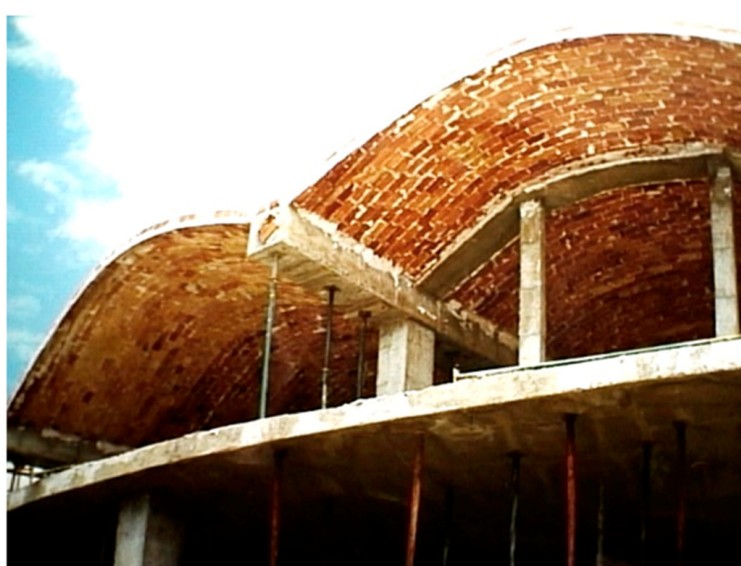

**Figure 27.** Vault pieces are constructed in hollow thin brick with steel mesh for reinforcement and attached to a concrete frame.

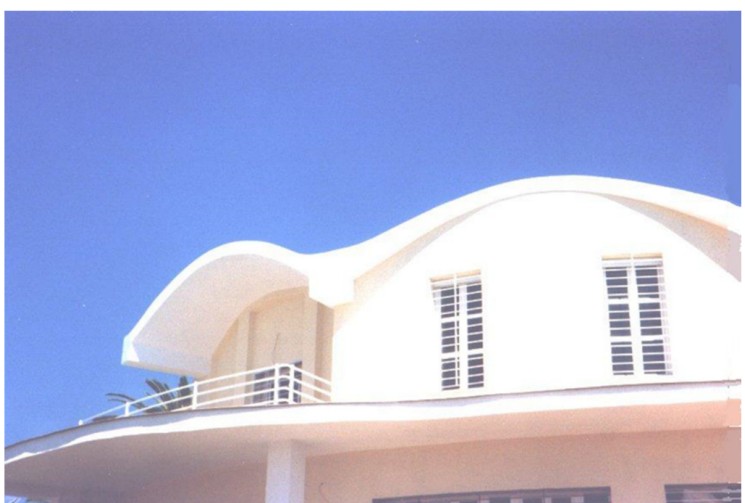

**Figure 28.** Detail of pseudo-conoid roofs in which the sun-path produces intriguing variations through the day.

### 5.3. Future Proposals

As a corollary to the theories elucidated, we will discuss two project-forms that we have created with conoids, taking into account evolving technologies. The first one is a system of skylights similar to the one depicted in Figures 21 and 24, but in this case, the glazed parts instead of being planar are also conoids (Figure 29). Bearing in mind the discussion on heat and light transfer of Section 3, this feature presents undoubted advantages [22]. Firstly, the glazing is better shaded and protected by the opaque upper conoid. Secondly, sunlight and heat transmission are modulated by the smooth curves conjoined to the innovative glass properties. In this way the glass surface becomes load-bearing and collaborates with the general structure. The form can be easily adapted to arrays of skylights (Figure 30).

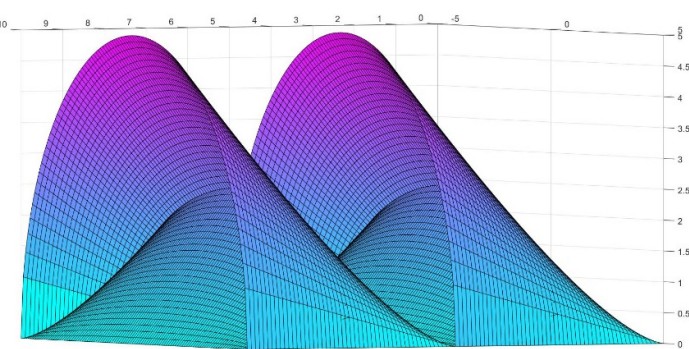

**Figure 29.** Outline of proposed skylights with internal conoid glazing.

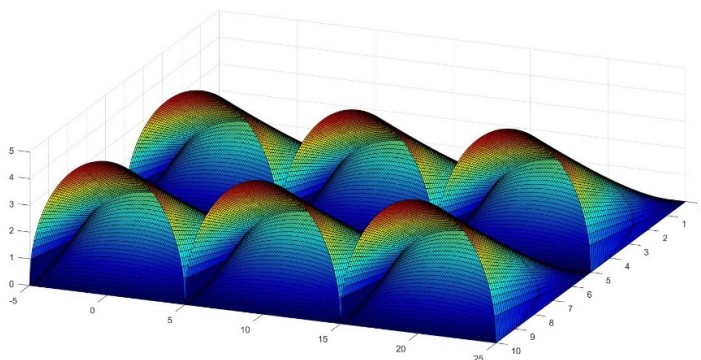

**Figure 30.** A suggested array of six conoidal skylights.

The new skylights are more impervious, break-proof, safer and cleaner in the absence of maintenance, as dust collection is diminished with the curvature.

The second form consists of an innovative proposal for an amphitheater, music or sports venue (Figure 31).

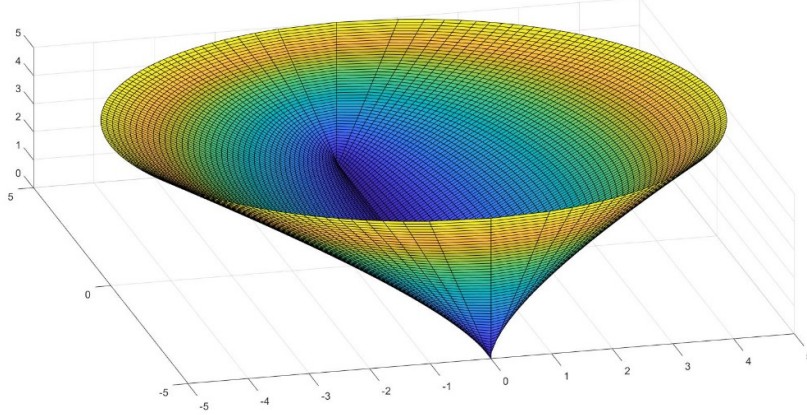

**Figure 31.** 3D drawing of the new Amphitheater.

In this case, being a double conoid, the advantages previously elucidated are even increased. The sections are closed curves, such as the ellipse and the circumference in the brim. Thus, they work as tension rings or girdles to hold the structure together without severe deformations. The bearing capacity of the shape is extreme. The tiers of the amphitheater are the obverse of the external façade; there is no need to superimpose a conical structure inside a cylinder such as in the Colosseum or in Spanish bull-rings. Among other problems, the ancient structures were forced to build giant discharge vaults and galleries, which transferred severe thrusts to the outer façade. As a result, we have calculated that savings in building materials of this proposed facility could be massive.

Still, the structure can be easily constructed with straight beam elements and reinforcements. The foundations are pointing to the soil as in a kind of arrow, which means that it will be very stable, safe and simple to develop.

The outer surface of the conoid is not vertical but inclined, and so the surroundings of the amphitheater would be self-shaded—an interesting feature in warm climates.

As for the grandstands, it is not difficult to adapt awnings or other shading systems to the inside area in order to protect the tiers from the rain or the sun (Figure 32).

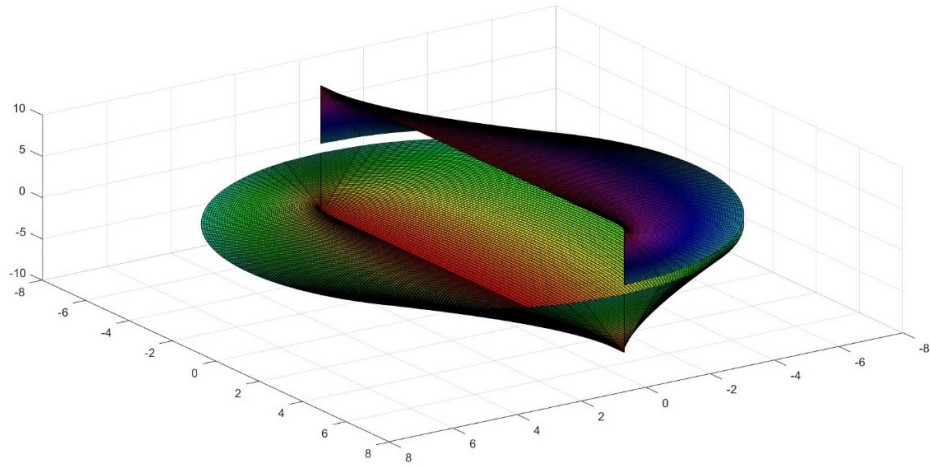

**Figure 32.** The Amphiteater with proposed retractable shade.

As mentioned above, the effect of concentration of sound, rather annoying in conventional stadiums will be almost completely avoided due to the convexity of the surface. In Reference [6], we have demonstrated such effect by acoustic ray-tracing procedures. These rely on the finding of the normal to the conoid surface at each point.

The procedure to extract the normal is first-order differentiation of the equation of the surface defined as $F(x,y,z)$. The normal vector is obtained as $N = (F_x, F_y, F_z)$ and in this case, has the value of:

$$N = \left( \frac{R^2 z^2}{(R-x)^3}, \; y, \; \frac{R^2 z}{(R-x)^2} \right) \tag{61}$$

We can trace a vector field with the reflected sound-rays from an emission point to check that they are effectively dispersed in the air and not concentrated [6].

The only remaining questions would be those of selecting the relative heights of the stages and platforms and other design issues such as circulation in the venue. Nonetheless, we believe that our proposal could be another good example of how the conoid-based bodies are able to create a significant volumetric space with a comparatively small enveloping surface; the key is that they offer a high spatial compactness, which is usually an added value in terms of heat exchange, costs reduction and sustainability, in general.

## 6. Repercussions for Technology

So far, we have presented scientific design developments that we hope will find a myriad of applications in technological areas such as Nautical, Aerospace, Building, Heritage, Retrofit and associated industries or machinery. Such facts attest to the versatility and feasibility of the solutions presented, which derive from our mathematical investigation. In the last part of the research, due to their complexity, the surfaces have been materialized with the help of 3D fabrication procedures to help decide about some difficult points of the equations or to reflect on future realizations of the proposals. It is undeniable for us that the results attain to the domains of Art and Design (Figures 33–36). They have been included in the study, as they constitute a veritable revolution of forms.

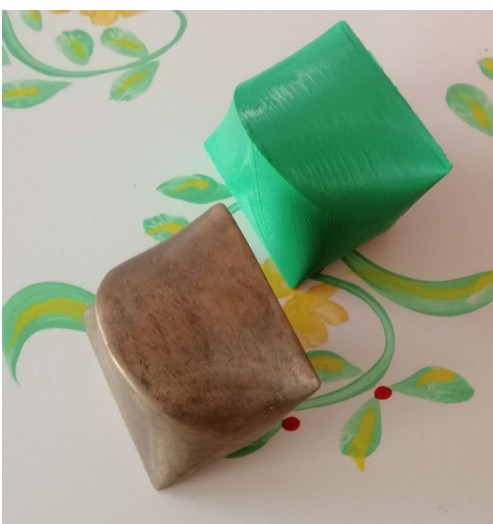

**Figure 33.** Antisphera in bronze by the artist Sergio Portela and in plastic 3D print (green).

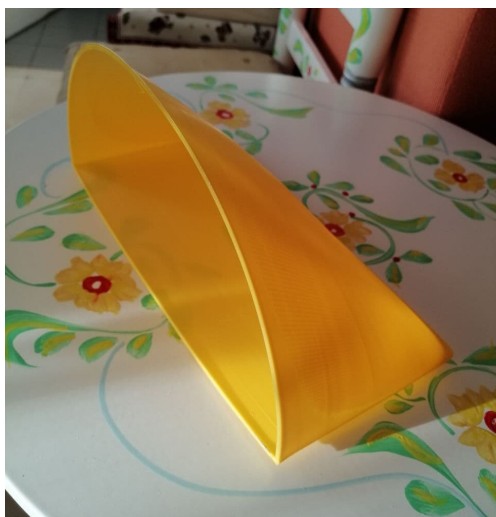

**Figure 34.** 3D print of a conoid similar to Doganoff's skylight.

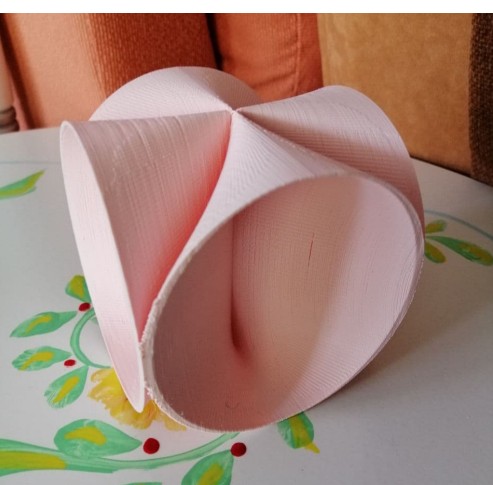

**Figure 35.** The Dyosphera in 3D print, detail of interlocks.

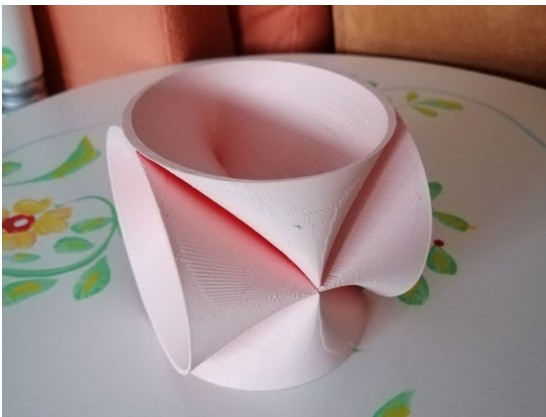

**Figure 36.** The Dyosphera standing on one of its four bases.

We believe that the implications of this geometrical advance are far-reaching. Due to its internal logic, it would be suitable for biotechnology. Especially the last development, the *Pterasphera*, being of a tubular nature, would be prone to fluid transportation. As a spring-like configuration it conjoins flexibility and balance. In Figures 37 and 38, we present examples of possible association and growth in parallel or opposed patterns.

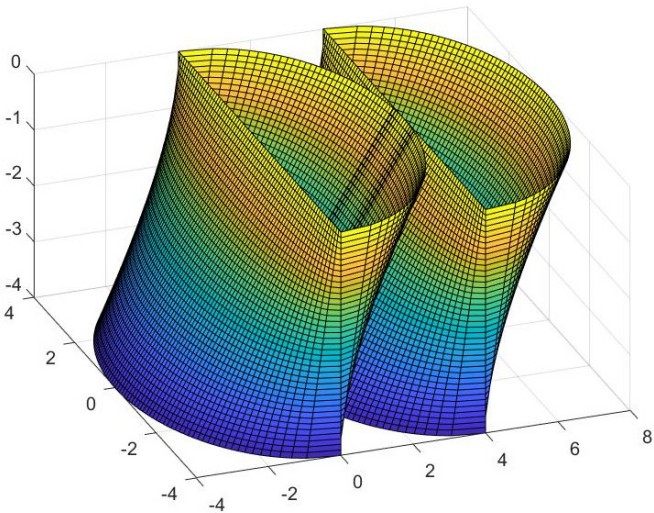

**Figure 37.** Two parallel *Pterasphera* tubules.

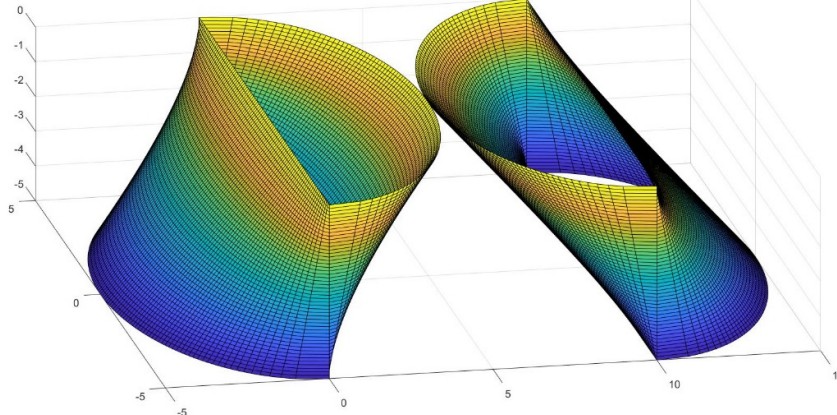

**Figure 38.** The same tubules opposed.

If we analyze the internal section of the tubules (Figure 39), it is composed of two semi-ellipses of varying sizes, but the span is constant at *R*; the extreme one is a semi-circumference of radius *R* and the middle horizontal section is a complete ellipse of minor axis *R* and major axis 2*R*.

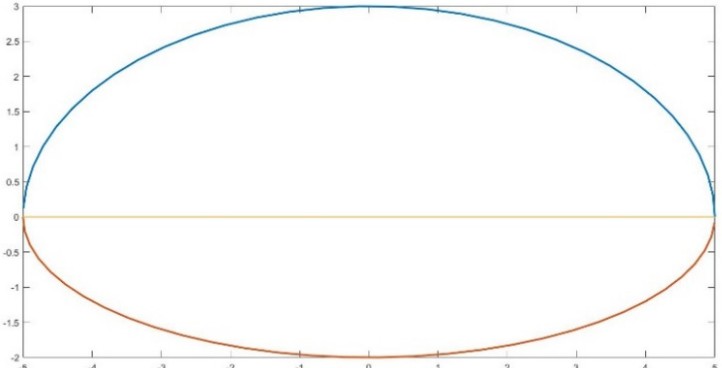

**Figure 39.** Horizontal section of a *Pterasphera* of major axis 10 and minor 5.

The dimensions of the section are constant, but its shape is not; thus, the velocity of the fluid inside the tubule can be deftly regulated from the same form. This would offer a clear alternative to reduce the noise level in the ducts or to decant particles in suspension.

The longitudinal central section of the *Pterasphera* gives a lozenge of horizontal dimensions R and the sides are inclined to the angle $(\pi/2 - \theta)$, with $\theta = \arctan(L/R)$. The perpendicular distance between inclined sides is of $d = R\cos\theta$.

This characteristic will facilitate insertion in existing rectangular ducts. for example, in retrofits. Mass fabrication is also simple since the form allows cuboid molds of the middle section over $2R \times R\cos\theta$.

In Figures 40 and 41, we present examples of vertical growth of the tubules, resembling vegetal pillars. Such form connects with the art tradition of coloana infinutului by Constantin Brâncuşi and with the architectural orders of classical architecture.

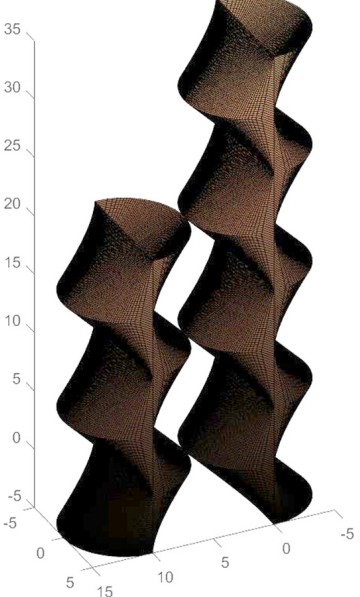

**Figure 40.** A pair of tubules of different height.

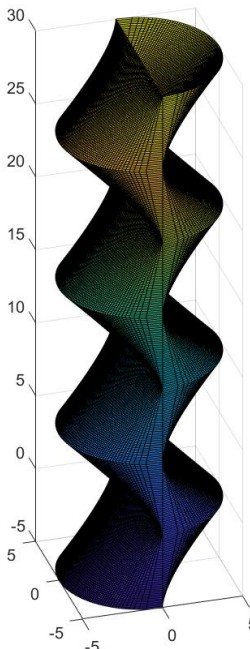

**Figure 41.** Different perspectives of the tubule.

We decided to explicate this finding because of the possibilities that it showcases, for instance, in tower buildings. Vertical connections are always feasible at the middle plane of the column, but the external envelop will benefit from the sun-tracking or shading properties already elucidated, combined with new photic materials of variable transparency (Figure 41). In consequence, lighting, thermal and acoustic features will considerably improve the existing conditions.

The structure of the so-conceived tower can be as lightweight as desired due to its inner balance and counterweight. As we have explained previously, alone or better in groups (Figure 42), it should perform adequately under earthquakes, extreme wind conditions or other unpredictable circumstances.

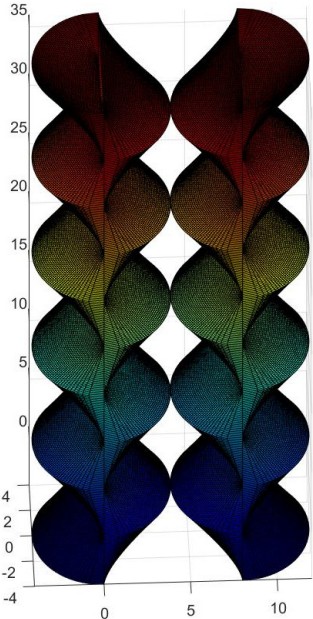

**Figure 42.** A symmetric array of twin tubules.

Conversely, the authors, by designing operations have achieved the form of a straight tubule of reversed sections.

Diverse technological fields could have a keen interest in the new forms that we have found, both for the macro- and micro-scales. It is indeed a leap into a sustainable future, which we hope go beyond.

## 7. Conclusions and Future Aims

In the first part of the article, we have identified adroit mathematic procedures, which greatly enhance our understanding of conoidal shapes. In the process, a new transcendental number has been obtained. We would call this cipher $\psi$. Being of a very precise nature, it provides a surprisingly accurate approximation of $\pi$. Thanks to Ramanujan's conjecture, we have been able to find the lateral area of the conoid, a recurrent form in organic structures whose scientific and technical knowledge was insufficient, although much desired for the benefit of art and architecture.

With such a procedure, we have created no less than four new types of figures that present high potential in many realms, such as aerospace, naval industries, transport, communication, biotechnology and fluid, light and sound-conducting devices.

Consequently, we have developed a vast array of revolutionary forms that showcases their utility for the design of sundry aeronautical parts and vessels. Due to their particular geometric properties, they can work, both for heat storage or dissipation, as the case may be. They perform aptly in thermal, luminous and acoustic radiation domains.

As a concession to the architectural discourse and paradigm, we have discussed how the notions that led to this form have evolved since early modern times; former examples of structures demonstrate integration of tradition with sustainability.

Further advantages to be outlined include that they enhance the employment of engineering and architectural forms, which can save considerable amounts of materials and building time, not only when used as cover or roofing in the sense of vaults or domes, but also vertically for cantilevered stadiums and amphitheaters, based on the cylinder since the Roman colosseum. Additionally, towers and high-rise buildings can be designed after the last form presented, *Pterasphera*, owing to its tubular nature, which opens the way for its use in transport infrastructures.

From the building perspective, easy reinforcement for pre-stressed construction and enhanced structural behavior could be other reasons for their use. Self-shading of the structure and built-in protection from weather phenomena are added values and contribute to its durability. An ample field of research in new materials is inaugurated with the properties revealed by the said forms.

The repertoire of possibilities derived from such geometric findings and explorations seem to be never ending.

Finally, from the mathematical point of view, a precise and non-trivial meaning has been attributed to the number $\pi^2$; this is an unexpected achievement that leaves us room to speculate on the notions of $\pi^3$, $\pi^N$ and $\pi$ elevated to infinite power $\infty$ and their applications in future art and architecture.

**Funding:** This research received no external funding.

**Institutional Review Board Statement:** Not applicable.

**Informed Consent Statement:** Not applicable.

**Data Availability Statement:** Not applicable.

**Acknowledgments:** This article is dedicated to J. M. Cabeza Arroyo. The author would also like to express remembrance for the late J. R. Perez de Lama who showed great interest in the early phases of this research. J. L. Perez de Lama Halcón was helpful in the preliminary 3D fabrication models used by the author to ascertain the hypotheses. J.C.-L. wants to thank Francisco Salguero-Andujar for his collaboration in checking the results of column 4 in Table 1.

**Conflicts of Interest:** The author declares no conflict of interest.

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
