# Peer review of "Architectural Characteristics of Different Configurations Based on New Geometric Determinations for the Conoid"

_buildings, doi:10.3390/buildings12010010_

Round 1
Reviewer 1 Report
To achieve the objectives, the authors identified the new way of describing the conoidal geometries using the new number ψ, which closely refers to the number Pi. The authors present different conoidal geometries, Antisphera, Dyosphera, Alopshera, Pterasphera and their combinations.
For completeness, the paper identifies the significance of finding the new number ψ, describing the new geometries. Still, the challenge lies in using those complex geometries in architectural and engineering solutions. The authors demonstrate the effective use of computer-aided tools to describe complex geometries but present architectural designs with conoidal shapes that cover only the simplest types of conoids. The paper covers a lot of points and maybe because of that it's rather blurry than easy to understand.
A few things are still missing or need improvement:
- a standard procedure to design a roof structure should be given, including all the factors in each stage of the process (geometrical, architectural, engineering), if readers want to design their product.
- The author presents the easiness of building conoidal structures. Is the conoid structurally efficient in terms of stresses, buckling? How do materials influence the conoids structural optimization and assembling?
- State of the art section is incomplete with the literature reviews. The authors presented the data as quite historical. What about more contemporary research on this topic? Are the conoids still the research topic?
- To understand all the key points and benefits of conoid, maybe it would benefit the paper's clarity, to at the flowchart demonstrating the factors of designing conoidal roofs?
- What are the characteristics of the roofs that may affect the choice of one of the project forms? What are the limitations in using those conoidal shaped roofs? What are the relations of curved glazing surfaces in terms of sustainable material usage? Also presented 3d printed models indicate that 3d printing technology is not yet adjusted to architectural scale, which materials authors consider as the future materials for such designing.
- How computer-aided designing might improve designing conoidal shapes? Since most of the presented architectural designs are quite old, what is happening in this pattern nowadays?
- What are the limitations of further studies?
- Citations should not be included in the 'Conclusion and future aims' part. The author(s) should provide a critical evaluation of the data provided, highlighting the advances that have been made and where the field is heading.
- Also, add the objectives/research questions of the study.
Abstract: The abstract would benefit from the sharper formulation of studies/problem/question, problem/answer question and implication of research results in a broader context of architectural engineering (who will benefit from knowing what the authors discovered in their research).

Author Response
We want to thank the reviewers for their kind comments and keen interest in our work. In the first place I want to make clear that we submitted our manuscript to a Special Issue of the Buildings Journal devoted to:
Special Issue "Architecture: Integration of Art and Engineering"
Some of the keywords of this issue, mentioned in the web page are:
- Art
- Construction engineering
- Cubature architecture
- Interior architecture
- Process design
- Design paradigms
For instance, Art cannot be defined in strict scientific terms. That is the main reason why in our article, as has been justly remarked by the reviewers, while trying to produce something new related with art or design, in some aspects of the discussion may present uncertainties from a stricter scientific perspective.
Regarding the comments in detail:
A few things are still missing or need improvement:
- a standard procedure to design a roof structure should be given, including all the factors in each stage of the process (geometrical, architectural, engineering), if readers want to design their product.
Answer: This paper is about creating new possible shapes for Art and Architecture, but it tries to do so in a scientific manner when possible. That said, our intention is to produce a variety of designs and products which transcend architecture and building construction and may attain aeronautics, aerospace, naval design and other related or auxiliary industries. Design of roof structures is a by-product in our paper, but still it has been presented for two main reasons: 1.- The authors have designed and constructed a number of innovative roofs following the methods hereby exposed and so their own projects may serve as a demonstration of the real potential of their findings. 2.- Design of shell roofs is very evident for the public and we believe that it definitely beckons the interest of the readers.
Focusing in the design of roof structures, the whole process has been integrated by parametric design in the Matlab interface. Thanks to its coherence, once the fundamental mathematic and geometric determinations are fulfilled, the results can be incorporated to a BIM (building information modelling) framework which, in turn, produces the necessary drawings, instructions for the builders and forms for legal regulations. Also, in a semi-automated manner, the BIM, yields a document ready for the tenders and executive project. With all the former, the structure can be realized with perfect ease until completion.
- The author presents the easiness of building conoidal structures. Is the conoid structurally efficient in terms of stresses, buckling? How do materials influence the conoids structural optimization and assembling?
Answer: As previously mentioned, our main objective is not structural calculation or even building some evolved conoidal structures. However, since the conoidal shape is formed and based on straight lines, whe have experienced that most kinds of prefabrications or in-situ preparations are well suited. In fact, in our limited experience for the proposals that we have calculated and built, reflected in the article, the structure has proved very efficient against buckling, bending and other stresses. This is in part a result of the geometry composed of rings (ellipses) and poles (lines), where the poles provide stability and the rings act as “girths” , giving flexibility against unforeseen effects due to the loads induced. Depending on the type of project a variety of materials can be used. Reinforced concrete is especially adequate and utmost economic for its use in conoidal shapes as Doganov’s project in Bulgaria attests. Steel or metal shells can be used to advantage in other cases. The present author has used steel beams and even reinforced ceramics completing the ideas of cohesive construction of Guastavino in the US and Eladio Dieste in South America, among other prominent engineers. In the future we believe that laminated composites and other plastics and fibres can be gradually incorporated to advantage.
- State of the art section is incomplete with the literature reviews. The authors presented the data as quite historical. What about more contemporary research on this topic? Are the conoids still the research topic?
Answer: Being an article on art and architecture, we find it natural to focus on historic and heritage examples. Regarding the contemporary aspects, as we have exposed ,there is very little or no contemporary research on the matter, perhaps only our own publications. that is the main reason why we decided to produce this pioneer work that may help to revitalize such interesting forms under a new and more scientific perspective.
However, we have identified from the beginning, the two following articles that mention the conoid from a structural point of view and we have added them to the references. They are interesting but do not deal with the geometry, the area or the volume of the conoids, only with the type of finite elements that can be applied to study the form in a stress-strain point of view. More specifically they feature finite elements consisting of two straight parallel borders and two differently curved parallel borders, and not the figure as a whole. Besides, the type of conoids that these authors study are not circular straight conoids as the ones that we discuss in our article.
- Das H.S., Chakravorty D. Design aids and selection guidelines for composite conoidal shell roofs—A finite element application. Reinf. Plast. Compos. 2007;26doi: 10.1177/0731684407081380.
- Sarmila Sahoo, "Dynamic Characters of Stiffened Composite Conoidal Shell Roofs with Cutouts: Design Aids and Selection Guidelines", Journal of Engineering, vol. 2013, Article ID 230120, 18 pages, 2013. https://doi.org/10.1155/2013/230120
In a recent article referenced below, there is an interesting study on composite materials and their hygrothermal properties when used in a conoid. Again this particular geometry in the article is just nominal and not studied in deep.
3.- Chaubey, A.; Kumar, A.; Fic, S.; Barnat-Hunek, D.; Sadowska-Buraczewska, B. Hygrothermal Analysis of Laminated Composite Skew Conoids. Materials 2019, 12, 225. https://doi.org/10.3390/ma12020225
Finally, the following interesting article does mention the parametric design of roof shells that can be conoids in certain circumstances, but not from a mathematic point of view and solely focusing in design software like Grasshopper or Karamba
4.- Dzwierzynska, J.; Prokopska, A. Pre-Rationalized Parametric Designing of Roof Shells Formed by Repetitive Modules of Catalan Surfaces. Symmetry 2018, 10, 105. https://doi.org/10.3390/sym10040105
Therefore, we conclude that there is almost no contemporary research on this form from the point of view of mathematical geometry and less than that as applied to art and architecture.
- To understand all the key points and benefits of conoid, maybe it would benefit the paper's clarity, to at the flowchart demonstrating the factors of designing conoidal roofs?
Answer: Yes, we have considered this possibility of the flow chart, but as we have informed previously in an article about art a flow chart could be deemed excessive for the reader. Besides, the techniques described in our article, are not limited to roofs, they are principally encouraged for towers and tall buildings, stadia, concert halls, temples, fountains and all forms of ornamental and landscape art. They could be applied not only to the macroscale but to the microscale as well.
Even so, we believe that the explanation included now in the paper as a response to question 1 (see above) of the reviewers, may suffice to increase the clarity of said building procedures in a more satisfactory manner.
- What are the characteristics of the roofs that may affect the choice of one of the project forms? What are the limitations in using those conoidal shaped roofs? What are the relations of curved glazing surfaces in terms of sustainable material usage? Also presented 3d printed models indicate that 3d printing technology is not yet adjusted to architectural scale, which materials authors consider as the future materials for such designing.
Answer: As exposed elsewhere, the article’s main objective is to offer scientific support for art and architecture. Regarding the specific question, the scale and requisites of the project clearly affect the choice and scope of the form.
The main limitations of conoidal roofs might be sometimes budgetary, although in our experience they are usually cheaper than flat slabs. Other times there might not be enough space or room for using this form. Different climates may introduce unexpected issues. But generally speaking, our intention is that the employ of this form as a roof is virtually boundless and that is one of its proposed advantages.
Curved shapes for glazing always cover more volume for less surface area. They are very effective in the production of architectural space. Therefore, if conveniently designed they are well suited for architectural design which as we know, has to do with the creation of as much liveable space as possible.
Regarding the capture of light and solar radiation, the curved shape of glazing is much more effective for this purpose and it comes with less side-effects like excessive warming which is usually colligated with typical plane surfaces and plates. From the aerodynamic or ventilation point of view we can say something similar. Acoustic advantages of the curved glazing are very remarkable as there are less echoes and undesired reflections or noise.
It is understood that at the usual skylight scale, curvature of the glazing is generally achieved by purposely orienting transparent tiles or tessellation conveniently framed with mastics and rubber joints and not by curving the glass panes themselves, which can be expensive and cumbersome for the time being, albeit future developments are expected in this sense.
Regarding the question of 3D printing, yes this technology has not as yet reached the standards of Architecture. But, as previously mentioned, there are several suitable materials like reinforced concrete, metal shells and reinforced ceramics available. The future for these designs clearly belongs to laminated composites and perhaps carbon fibres, quite pricey nowadays.
- How computer-aided designing might improve designing conoidal shapes? Since most of the presented architectural designs are quite old, what is happening in this pattern nowadays?
Answer: Computer Aided Design understood in the precise way practised in this article, with the help of the scientific formulations developed and introduced in the Matlab software is actually contributing to reveal the hidden potential of conoidal shapes and make them truly available for an ever-increasing range of products and design problems. Before our findings, its use was much reduced and perhaps limited to the authors’ own architecture. From now on, we firmly believe that its use will become massive and not exclusively subject to the boundaries of architecture.
- What are the limitations of further studies?
Answer: We consider that after our article there remain very few limitations on further studies. Perhaps more experiments with the built elements could be required and might be difficult to obtain. More information about financial costs could be required and also post-occupancy evaluation. The adverse effects experienced in harsh climates might be also a factor.
- Citations should not be included in the 'Conclusion and future aims' part. The author(s) should provide a critical evaluation of the data provided, highlighting the advances that have been made and where the field is heading.
Answer: citations have been removed from the conclusions. We have now given a critical evaluation of the data provided and highlighted the advances and the possible direction of the field.
- Also, add the objectives/research questions of the study.
Answer: the objective of this article is to orient and enhance the evolution of new architectural and artistic forms offering the most updated scientific and mathematic support to this aim. We expect to create a true revolution of forms.
Abstract: The abstract would benefit from the sharper formulation of studies/problem/question, problem/answer question and implication of research results in a broader context of architectural engineering (who will benefit from knowing what the authors discovered in their research).
Answer: The abstract has been formulated in a sharper way to stress the questions, the process and the findings in the field of architectural engineering. The potential benefits of the findings presented are shown in a clearer fashion.

Reviewer 2 Report
To see the formulas and figures please find in the attached pdffile.
The properties of the special (two rectilinear components and the segmental directrice lie in a horizontal plane, the circular directrice – in a vertical plane) circular conoid were discussed in the paper. Each, in fact, an elliptical conoid is a circular one. The group (sum) of two (or four) circular (or semicircular) conoids, whose rectangular projection is a square, the author called the “Antisphera”. It is an interesting analog of a sphere with respect to some shape of a circular conoid. Some properties seem to be applicable in architectural (and not only) design. In the opinion of the reviewer, it is worth reading the comments below.
Notes and comments:
1: The circular conoid surface equation (1), (2) is easy to solve for any variable. Therefore
↔ = or = .
Hence, theoretically (in order to compare the results of the work), it is easy to obtain the classic integral formula (surface integral of the first type) for the surface area and the flux formula of the vector field (surface integral of the second type - the matter of radiation). Of course, it is not necessarily easy to accurately compute these integrals. This approach allows, for example, examining the issue of covering thickness in terms of assessing the thickness of the insulation layer.
2: In the line 32: instead of „… generated by parallel straight lines…” should be „… generated by straight lines parallel to one fixed plane...”. In fact, these straight lines are skew.
3: In the line 69, according to the previously adopted coordinate system (rules (1), (2) and Figures 1, 2), instead of = (Figure R2) should be (see Figure R1), unless we write the conoid equation as + = but then we have to change the Figures 1, 2.
|
+ = |
+ = |
|
= |
= |
|
|
|
|
Figure R1: |
Figure R2: |
This fact does not affect the value of the integral (10) (line 97), after the substitution u=L-x (du=-dx, x=0, u=L; x=L, u=0) we get the same integral ( ).
4: The formula (3) between lines 63-64 is not an exact formula, hence the number ψ has an unknown error. It would be worthwhile to provide this error or at least to make a constructive comment on this fact.
5: Figures 25 and 26 show a cover that does not consist of a conoid in the sense discussed by the author. In this case, the elements of the cover are the so-called “cylindroids”. The projections of the forming directrices are not straight lines parallel to any fixed plane.
6: The potential uses of the conoidal surfaces in the applications have not been proven. These are only visual architectural analyzes. Other insights are the author's loose expressions. No sufficient analyzes have been provided in terms of building mechanics, fluid mechanics, aeromechanics, ...
7: If the Editorial Office deems it appropriate, some figures can be omitted in the paper. Examples of architectural realizations are not directly related to the paper results, while the visualization of solids derived from the conoid can also be limited.

Author Response
We want to thank the reviewers for their kind comments and keen interest in our work. In the first place we want to make clear that we submitted our manuscript to a Special Issue of the Buildings Journal devoted to:
Special Issue "Architecture: Integration of Art and Engineering"
Some of the keywords of this issue, mentioned in the web page are:
- Art
- Construction engineering
- Cubature architecture
- Interior architecture
- Process design
- Design paradigms
For instance, Art cannot be defined in strict scientific terms. That is the main reason why in our article, as has been justly remarked by the reviewers, while trying to produce something new related with art or design, in some aspects of the discussion may present uncertainties from a stricter scientific perspective.
1: The circular conoid surface equation (1), (2) is easy to solve for any variable. Therefore ? 2 = 1 ? 2 (? 2 − ? 2 )(? − ?) 2 ↔ ? = 1 ? |? − ?|√(?2 − ? 2) or ? = − 1 ? |? − ?|√(?2 − ? 2) . Hence, theoretically (in order to compare the results of the work), it is easy to obtain the classic integral formula (surface integral of the first type) for the surface area and the flux formula of the vector field (surface integral of the second type - the matter of radiation). Of course, it is not necessarily easy to accurately compute these integrals. This approach allows, for example, examining the issue of covering thickness in terms of assessing the thickness of the insulation layer.
Answer: We thank the reviewer for making this suggestion. Based on it, we have clarified in the abstract and the introduction that the regular ways of finding curved areas by means of differential geometry, despite their complex development, do not yield any applicable results or at least the present authors and the experts or colleagues consulted at the mathematics departments of the university, have been unable to find a feasible solution for the last fifteen years. That is why the authors devised another procedure not deprived of serendipity, since it is based on the Ramanujan approximation, that gives an accurate although not exact result. Such result is deemed sufficient for engineering mathematics and architectural engineering purposes and gives enough way for further developments.
2: In the line 32: instead of „… generated by parallel straight lines…” should be „… generated by straight lines parallel to one fixed plane...”. In fact, these straight lines are skew.
Answer: Yes, this is quite right and has been corrected.
3: In the line 69, according to the previously adopted coordinate system (rules (1), (2) and Figures 1, 2), instead of ? = ?·? ? (Figure R2) should be ? = (?−?)? ? (see Figure R1), unless we write the conoid equation as ? 2? 2 ? 2 +? 2 =? 2 , but then we have to change the Figures 1, 2
Answer: We have now explained at the beginning of the operations that the x-axis is reversed in direction with respect to figure 2 and that the origin of coordinates is placed for convenience’s sake at the linear extreme of the conoid.
4.- The formula (3) between lines 63-64 is not an exact formula, hence the number ψ has an unknown error. It would be worthwhile to provide this error or at least to make a constructive comment on this fact.
Answer: yes, this comment seems opportune and has been added in the corrected version. The error is not constant since it depends on the diverse conoidal sections. Ramanujan’s expression number two has been especially selected because two reputed authors quoted in the references, perceive in it the smallest error for flatter ellipses tending to a straight line. These kind of ellipses are characteristic of the regular circular conoid. On the other hand, as there is no exact expression for the length of an ellipse, there is no proper way to compute the error in a mathematical fashion. However, the overall error for the calculation of the surface, has been estimated in table 1 by means of numeric methods incorporated respectively in the Matlab and Rhinoceros software packages.
5: Figures 25 and 26 show a cover that does not consist of a conoid in the sense discussed by the author. In this case, the elements of the cover are the so-called “cylindroids”. The projections of the forming directrices are not straight lines parallel to any fixed plane.
Answer: yes, indeed, in these surfaces the forming straight lines are not parallel to a fixed plane but they converge along a vertical axis of symmetry. The authors have demonstrated it at a separate mathematical procedure that exceeds the engineering purposes of the manuscript. The new surface that arises has been termed Cabeza-Abajo surface after the authors. However, the authors admit that their inspiration for the new surfaces stems from the conoid.
6: The potential uses of the conoidal surfaces in the applications have not been proven. These are only visual architectural analyzes. Other insights are the author's loose expressions. No sufficient analyzes have been provided in terms of building mechanics, fluid mechanics, aeromechanics, ...
Answer. From the point of view of building mechanics and building aerodynamics, such properties have been proven indirectly by the excellent performance shown at the authors’ own constructions and edifices and at other still extant buildings that we have presented and discussed briefly in the paper. This is the main reason to include them in the figures, though they may seem not directly related with all parts of the theories exposed. A more complete fluid mechanics analysis is on its way in the laboratories of the university and we hope to present the results soon. However, the mathematic intuitions presented in the paper more than suggest an adequate or near optimal behaviour, especially the discussion on the normal, section and the easy construction demonstrated by a figure formed by straight lines and the linear equilibrium of forces that it provides.
7: If the Editorial Office deems it appropriate, some figures can be omitted in the paper. Examples of architectural realizations are not directly related to the paper results, while the visualization of solids derived from the conoid can also be limited.
Answer. Indeed, any figure can be limited at the demand of the Editorial Office. We have perhaps slightly increased the number of figures in an effort to demonstrate the feasibility and strong potential for Art and Architecture of the figures deducted and created from the theories that issue in the article.

Reviewer 3 Report
The paper addresses a very specific topic with a multidisciplinary approach. It starts from demonstrating how to possibly calculate the surface area of a conoid and flows through some possible applications of obtained results.
I strongly suggest changing figures that do not allow the correct reading of proportions with better proportioned ones, i.e., Figure 2 is shortened on y axis and can be changed with a proper isometric view. Same for Figures 6, 7, 8, 9, 10, 11, 12, 13. Moreover, Figure 10 is the same as Figure 3: I suggest removing it.
Since the paper presents clear multidisciplinary approach and the Author cites Guarini’s studies on approximation of the conoid volume, I think it would be interesting to compare Author and Guarini’s results.
I found few minor issues with English language that I recall in the attached .pdf table.

Author Response
The paper addresses a very specific topic with a multidisciplinary approach. It starts from demonstrating how to possibly calculate the surface area of a conoid and flows through some possible applications of obtained results.
Answer: we thank these reviewers for their keen understanding of our proposal
I strongly suggest changing figures that do not allow the correct reading of proportions with better proportioned ones, i.e., Figure 2 is shortened on y axis and can be changed with a proper isometric view. Same for Figures 6, 7, 8, 9, 10, 11, 12, 13. Moreover, Figure 10 is the same as Figure 3: I suggest removing it.
Answer: Figure 2 has been enlarged in the said axis. Figures 6,7,8, 9, 10, 11, 12, 13 have been proportioned following the suggestion. Figure 10 is maintained after the corrections because we still find it interesting for the readers to offer a different point of view in order to describe a newly created and unusual surface.
Since the paper presents clear multidisciplinary approach and the Author cites Guarini’s studies on approximation of the conoid volume, I think it would be interesting to compare Author and Guarini’s results.
Answer: thank you for their observation. In fact, Guarini’s proposal for the volume of the conoid that he calls a cone with a line ending, is based on triangulation after Euclid (His book is titled Euclid’s follower or addict) and is exact. Ours, which employs a new integral procedure instead of triangulation is also exact and therefore it yields the same result as in Guarini’s case.
I found few minor issues with English language that I recall in the attached .pdf table.
Answer: Thank you for pointing these flaws, we have corrected them and added two new columns in Table 1.